# A potential mechanism for tetraspanin CD82-mediated regulation of EGFR

Elisa Lamottke , T Harma C Brondijk , Piet Gros

**Epidermal growth factor receptor (EGFR) regulates cell growth, differentiation, and migration through mechanisms of ligand binding and dimerization. Tetraspanin CD82 is known to interact with and regulate EGFR; however, the underlying molecular mechanisms are not clear. In this study, we used detergent-solubilized and detergent-purified EGFR-CD82 complexes and fusion proteins to characterize the interaction of EGFR with CD82 by size-exclusion chromatography and cryo-electron microscopy. Our data show that CD82 binds monomeric EGFR and dissociates from EGFR dimers. Congruently, less EGF is bound to EGFR in the presence of CD82, likely because of reduced EGFR dimerization. AlphaFold2 multimer predictions together with a 15 Å resolution cryo-EM density map support a curved-back conformation of EGFR with a putative interaction site between monomeric EGFR domain IV and the large extracellular loop of CD82. Together, our results support CD82 regulating EGFR function by hindering dimer formation and show CD82 dissociation from EGFR upon EGF-induced EGFR dimerization.**

## Introduction

The epidermal growth factor receptor (EGFR) is one of four members of the ErbB family of human receptor tyrosine kinases. This family of receptors regulates cell proliferation, differentiation, and migration (Yarden & Sliwkowski, 2001). EGFR activity is up-regulated in several cancers and is an often-used target for lung cancer therapy (Ayati et al, 2020). Upon extracellularly binding EGF, a 53–amino acid peptide, the receptor dimerizes bringing the intracellular kinase domains in close proximity, resulting in their activation. The tyrosine kinases phosphorylate the C-terminal tail of the dimer partner, triggering different signalling cascades (Yarden & Schlessinger, 1987a). Signalling is terminated by internalization of the active complex via clathrin-coated pits and subsequent degradation in lysosomes (Sorkin & Goh, 2009). EGFR is a heavily glycosylated, type I, single-pass membrane protein of 1,186 amino acids and contains four extracellular domains, besides its intracellular kinase and kinase substrate domain. The arrangement of the extracellular domains of the monomeric receptor is highly flexible, alternating between a tethered, inhibited conformation and an extended conformation (Kaplan et al, 2016). In the inhibited conformation, domains II and IV are interacting through β-hairpins in each domain (Schmitz et al, 2013). In the extended conformation, a high-affinity ligand-binding site is formed between domains I and III of the receptor (Ogiso et al, 2002). EGF binding stabilizes the extended conformation and promotes receptor dimerization. The EGFR–dimer interface consists of the two β-hairpins in domains II and IV that were interacting intramolecularly in the inhibited conformation followed by interactions in the transmembrane domains and additional interactions within the intracellular domain (Lu et al, 2010; Schmitz et al, 2013). However, single-molecule tracking and FRET studies indicate that EGFR in the extended conformation can form unstable dimers in the absence of a ligand (Gadella & Jovin, 1995; Sako et al, 2000; Hofman et al, 2010). At low EGF concentrations, the receptor can form higher order oligomers, which trigger signalling and internalization (Yarden & Schlessinger, 1987b; Hofman et al, 2010).

Tetraspanins are a membrane protein family defined by four transmembrane helices and a small extracellular loop and a large extracellular loop (LEL). The LEL contains four to eight cysteines and a 100% conserved CCG motif. Tetraspanins are involved in several cellular processes, including receptor regulation (Kummer et al, 2020), protein trafficking (Berditchevski & Odintsova, 2006), endo- and exocytosis (Andreu & Yáñez-Mó, 2014), and virus budding (Hantak et al, 2019). To regulate receptor function, it is proposed that tetraspanins organize into tetraspanin-enriched microdomains or nanodomains (TEMs) of ca. 120 nm in diameter (Hemler, 2005; Zuidscherwoude et al, 2015). TEMs are thought to be formed via molecular interactions between membrane proteins and lipids (Odintsova et al, 2000; Charrin et al, 2003; van Deventer et al, 2017). However, TEMs contain less than 10 copies of a receptor–tetraspanin pair (Zuidscherwoude et al, 2015). Recent structural studies of receptor–tetraspanin complexes highlight the heterogeneity of interaction sites and associated functions. The structure of a CD19-CD81 fusion protein reveals interactions

---

Structural Biochemistry, Bijvoet Centre for Biomolecular Research, Department of Chemistry, Faculty of Science, Utrecht University, Utrecht, Netherlands

Correspondence: P.Gros@uu.nl

between the extracellular domains (Susa et al, 2021). These interactions prevent CD19 from joining a larger signalling complex, the B-cell receptor complex (Mattila et al, 2013). Another example, the metalloproteinase ADAM10 interacts with six tetraspanins (all containing eight cysteines in the LEL) (Haining et al, 2012). The interacting tetraspanin regulates substrate specificity of ADAM10 by positioning of the proteinase domain (Jouannet et al, 2015; Lipper et al, 2023). The complexes of EWIF or EWI2 with CD9 on the other hand display a different interaction site. Here, the interaction primarily occurs between transmembrane domains, forming a 2:2 heterotetramer (Oosterheert et al, 2020; Umeda et al, 2020). These complexes might be important for protein sorting and exosome formation (Umeda et al, 2020).

CD82, also known as KAI-1 or TSPAN-27, is a 267–amino acid protein with three disulphide bonds and three N-glycosylation sites in the LEL (Dong et al, 1995; Wang et al, 2012). The interaction between EGFR and CD82 was first demonstrated using co-immunoprecipitation (Odintsova et al, 2000). Immunofluorescence microscopy on cells subsequently confirmed colocalization of EGFR and CD82 in clusters, presumably TEMs (Odintsova et al, 2003). Functionally, CD82 inhibits EGFR-induced cell proliferation in response to EGF (Odintsova et al, 2000). This is possibly caused by reduced dimer formation and faster signal termination by CD82 in EGFR-expressing cells (Odintsova et al, 2000, 2003). Immunoprecipitation studies suggest that EGF destabilizes the EGFR-CD82 complex (Odintsova et al, 2000), and a recent single-particle tracking study showed reduction of EGFR-CD82 cluster size (in response to EGF) but no dissociation of EGFR and CD82 (Sugiyama et al, 2023). The same study demonstrated that TEM-associated EGFR exhibits fast EGF binding, possibly because of a conformational change induced by CD82. In addition, the study shows that EGFR confinement in TEMs reduces the formation of signalling oligomers. Despite these observations that CD82 influences EGFR function, the molecular mechanisms responsible for the variable effects are not clear yet.

In this study, we performed biochemical analyses of detergent-purified EGFR-CD82 complexes. EGFR and CD82 were either co-expressed or expressed as a fusion protein to investigate complex formation. The complex was analysed using single-particle analysis (SPA) cryo-electron microscopy (EM). Furthermore, the effect of CD82 on EGF-binding affinity, the stability of the EGFR-EGF complex, and EGF-induced receptor dimerization was analysed.

# Results

## Co-expression of CD82 and EGFR

We investigated whether EGFR and CD82 can be purified as a detergent-solubilized complex. EGFR-His$_6$ and CD82-eGFP-StrepII$_3$ were co-expressed in 4 ml HEK293-E+ cells and purified via the StrepII$_3$-tag on CD82. The resulting SDS–PAGE gel showed a strong double band at 50 and 70 kD along with several unknown, faint bands (Fig 1A). The double band was identified as CD82-eGFP-StrepII$_3$ via the eGFP fluorescence in the SDS–PAGE gel (Fig 1B). CD82 is N-glycosylated at three sites located in the LEL, and the two

bands likely represent two different glycosylation species and have been observed as a double band in previous studies (Odintsova et al, 2000; Wang et al, 2012). No band was observed at a size of 150 kD, expected for EGFR-His$_6$. However, copurification of EGFR-His$_6$ was confirmed by an anti-His Western blot, which showed a single band at 150 kD (Fig 1C). To further investigate whether the EGFR-CD82 complex can be purified, we scaled up the co-expression of EGFR-His$_6$ and CD82-eGFP-StrepII$_3$. Size-exclusion chromatography (SEC) showed a sharp peak at 16 ml elution volume (EV) and a shoulder between 13 and 15 ml EV (Fig 1D). The SDS–PAGE gel of individual fractions indicated that CD82-eGFP-StrepII$_3$ was present in the 16 ml peak. The shoulder between 13 and 15 ml contained a 150-kD band, consistent with the observed size of EGFR-His$_6$, the CD82 double band, and several unspecified bands (Fig 1E). In conclusion, the complex of EGFR and CD82 was purified from overexpressing mammalian cells and separated by SEC. However, the complex was only purified with low purity and a yield of 0.1 mg/litre culture.

## Expression of the EGFR-CD82 fusion protein

We designed a fusion protein linking the EGFR ectodomains (domains I–IV) and the transmembrane (TM) helix (constituting residues 1–672) to CD82 via a 31–amino acid linker containing a TEV-cleavage site flanked by two 4 × GGS repeats (Fig 2A). The EGFR(I-TM)-CD82 fusion protein was expressed in small scale with a C-terminal eGFP-StrepII$_3$-tag. The purification yielded ~2 mg/litre culture of EGFR(I-TM)-CD82-eGFP-StrepII$_3$, a large increase compared with the yield when EGFR and CD82 were co-expressed with improved purity. SDS–PAGE gel showed a strong double band, as for CD82-StrepII$_3$, at 140 and 160 kD and some minor, unknown bands (Fig 2B). The double band was identified as the target protein using the eGFP fluorescence (Fig 2C). To analyse complex stability, we cleaved the fusion protein with TEV protease. After 2 h of cleavage on ice, the EGFR(I-TM)-CD82 band almost disappeared and the cleavage products were visible, indicating ~95% of cleavage (Fig 2B and C). To assess the stability of the complex, we performed fluorescent size-exclusion chromatography (F-SEC) monitoring the eGFP fluorescence. The chromatogram of uncleaved EGFR(I-TM)-CD82-StrepII$_3$ showed a sharp main peak at 9.5 ml EV and a shoulder between 8 and 9 ml EV, possibly caused by co-purified unknown proteins. After different times of cleavage with TEV protease, the main peak, corresponding to co-eluting EGFR(I-TM) and CD82-eGFP-StrepII$_3$, reduced and a new peak appeared at 11 ml EV, which corresponds to CD82-eGFP-StrepII$_3$ (Fig 2D). The amount of co-eluting EGFR(I-TM) and CD82-eGFP-StrepII$_3$ reduced (relatively) rapidly in the first 2 h and appeared to decrease slower after 2 h. After ~6 h, half of CD82-eGFP-StrepII$_3$ still co-eluted with EGFR(I-TM) at 9.5 ml EV (Fig 2D), 4 h after we expected near-complete cleavage by TEV protease. In conclusion, the fusion of EGFR(I-TM) and CD82 largely increased the yield and purity of the protein complex. We demonstrated that upon cleavage with TEV protease, CD82-eGFP-StrepII$_3$ co-eluted with EGFR(I-TM), whereas the complex was almost fully cleaved after 2 h; the half-time of dissociation of CD82-eGFP-StrepII$_3$ from EGFR(I-TM) was after 6 h.

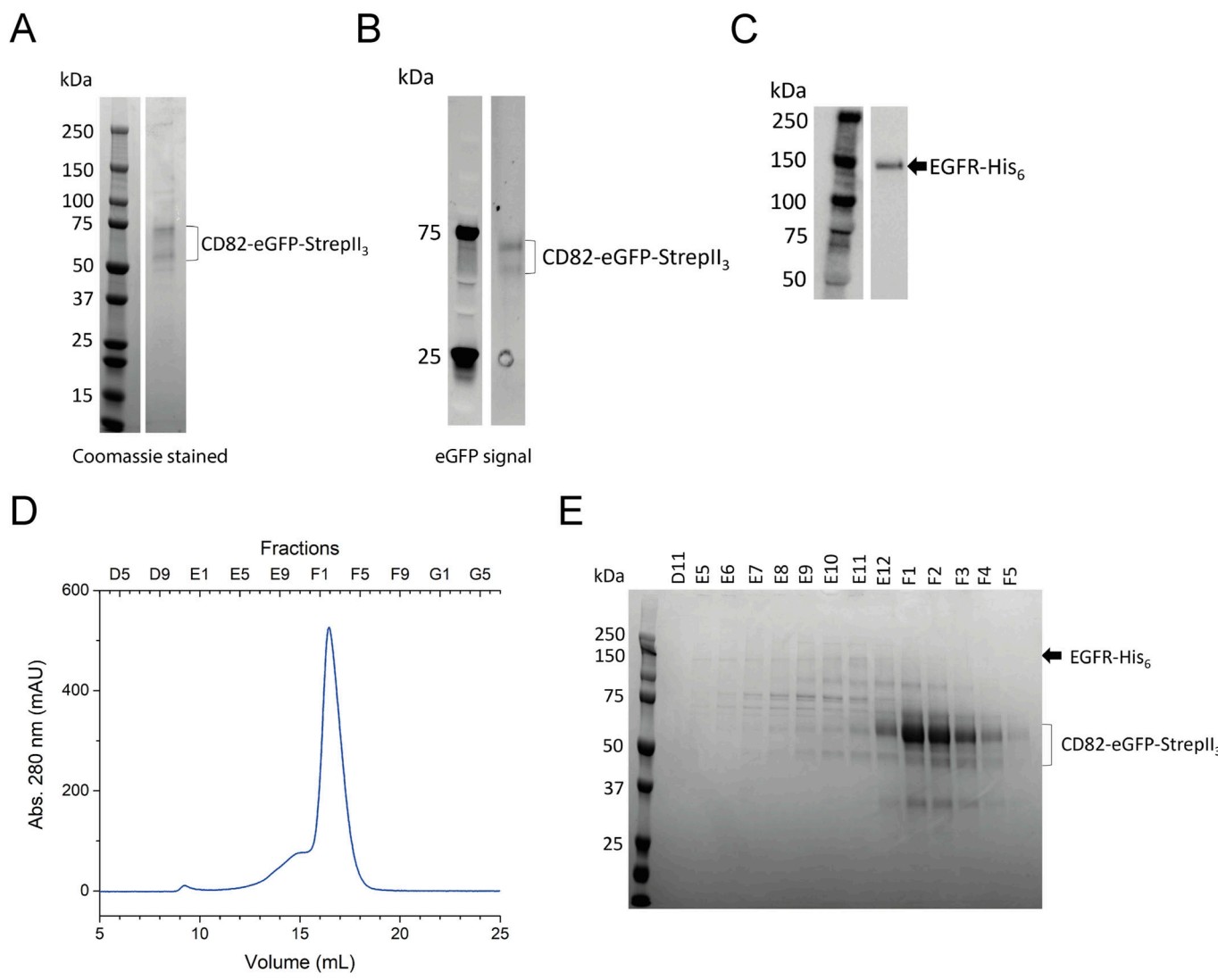

**Figure 1. Co-expression and purification of EGFR-His$_6$ and CD82-eGFP-StrepII$_3$.**
**(A, B)** 4–15% SDS–PAGE gel sample after Strep-Tactin purification stained with Coomassie (A) and eGFP fluorescence (B). **(C)** Anti-His Western blot of a 4–15% SDS–PAGE gel after Strep-Tactin purification. **(D)** SEC of large-scale purification using a Superose 6 Increase 10/300 Gl column. **(E)** 10% SDS–PAGE gel of single fractions from SEC. Source data are available for this figure.

### Cryo-EM analysis of the EGFR-CD82 fusion protein

To analyse the EGFR(I-TM)-CD82 complex further, we used SPA cryo-EM. Two samples of EGFR(I-TM)-CD82-StrepII$_3$ fusion protein were prepared, one with EGF-His$_6$ and one without. The SEC resulted in a single peak at 15 ml EV containing pure EGFR(I-TM)-CD82-StrepII$_3$ (Fig S1A and B). The fractions C6-C9 were concentrated to 5 mg/ml and plunge-frozen on EM grids. To analyse the influence of EGFR-ligand EGF on the complex, we added purified EGF-His$_6$ to the EGFR(I-TM)-CD82 fusion protein (Fig S2). The SEC showed an additional shoulder at earlier EV of 14 ml compared with the SEC without EGF-His$_6$ (Fig S1C). This shoulder might correspond to an EGF-induced dimer of EGFR(I-TM)-CD82-StrepII$_3$. The SDS–PAGE gel showed a pure protein co-eluting with EGF-His$_6$ indicating the formation of a stable complex between EGFR(I-TM)-

CD82-StrepII$_3$ and EGF-His$_6$, as reported previously (Fig S1D) (Odintsova et al, 2000; Sugiyama et al, 2023). The fractions C4 and C5 were concentrated to 3 mg/ml and plunge-frozen on EM grids.

We collected 3,133 movies of the EGFR(I-TM)-CD82 sample and 2,819 movies of the sample with added EGF-His$_6$ on a Talos Arctica (200 kV). The movies showed evenly distributed, homogeneous particles that were partly present in both data sets (Fig S3). A first subgroup, observed only in the presence of EGF, displayed a blurry micelle with a large, symmetric particle (Fig 3A). 2D averages of this subgroup strongly resembled the structure of an EGF-bound EGFR dimer (Huang et al, 2021). A second subgroup of 2D classes, present in both data sets, displayed a smaller, highly flexible averaged particle on top of a bigger micelle with at least two visible domains (Fig 3A). This suggested the presence of monomeric EGFR in these 2D classes (Lu et al, 2012). A third and last subgroup of 2D classes,

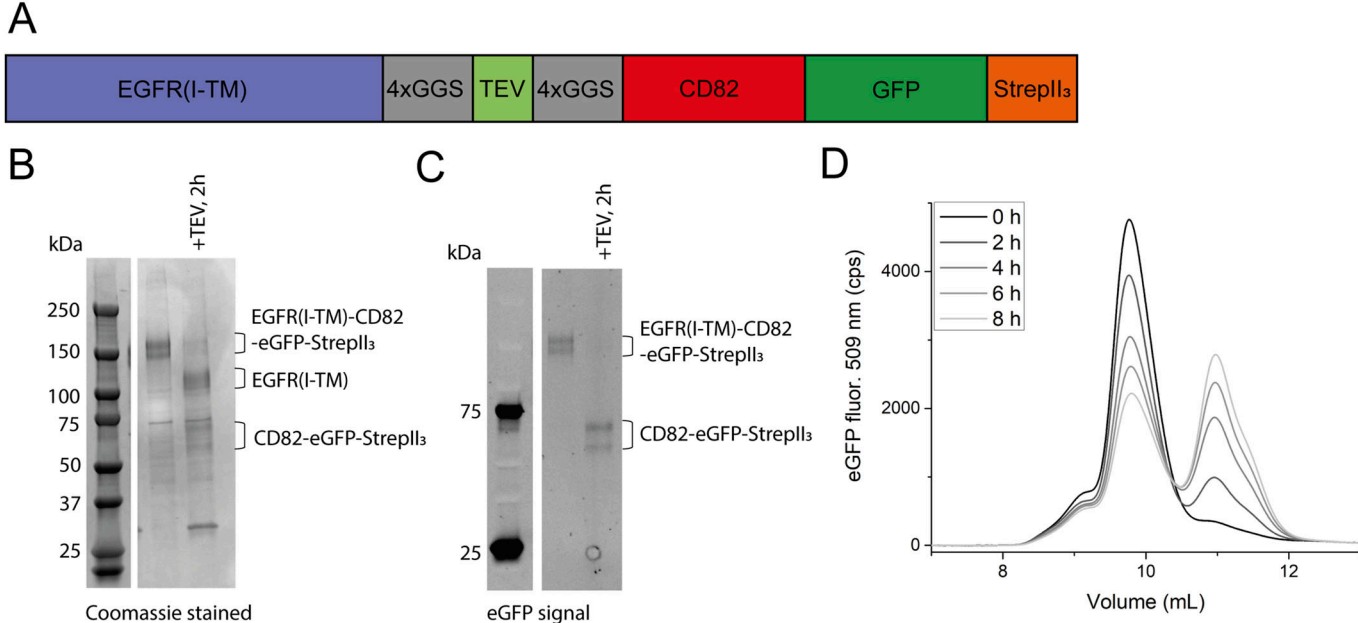

**Figure 2. Purification of the EGFR(I-TM)-CD82-eGFP-StrepII₃ fusion protein and complex stability.**
**(A)** Design of the fusion protein of EGFR and CD82. **(B, C)** 4–15% SDS–PAGE gel of small-scale purification of EGFR(I-TM)-CD82-StrepII₃ before and after cleavage with TEV protease for 2 h on ice stained with Coomassie (B) and eGFP fluorescence (C). **(D)** F-SEC of EGFR(I-TM)-CD82-eGFP-StrepII₃ at different time points after cleavage with TEV protease on a Superdex 200 Increase 10/300 Gl column.
Source data are available for this figure.

observed in both data sets, displayed a sharply resolved, slightly curved micelle containing an asymmetrically positioned particle of the expected size of a tetraspanin (Fig 3A). In some 2D classes, additional, blurry density is visible, suggesting the presence of EGFR that is highly flexible. For the 3D construction, particles were split in 2D classes showing dimerized EGFR and the other two subgroups. Both collected data sets showed 2D classes with two micelles in close proximity to each other that were not included in the first ab initio modelling (Fig S4A). The structure determination of EGFR dimers resulted in a density map with 6.5 Å resolution (averaged over 56,818 particles) showing a blurred micelle with an EGF-bound EGFR dimer with some resolved N-glycans (Fig 3B). Local refinements, excluding the micelle, resulted in a 5.7 Å density map that was in agreement with the previously solved structure of the EGF-bound EGFR dimer (PDB: 7SYD) (Huang et al, 2021) (Fig S5). No density resembling CD82 was visible, and the resulting micelle was too small to contain the transmembrane helices of two additional CD82 molecules. To further confirm the absence of CD82, we did additional local refinements using a mask around the micelle and EGFR(IV) and a set maximum resolution of 10 Å to avoid the loss of low resolution or flexible features. These additional refinements did not show additional protein densities (Fig S4B). In addition, a 3D volume resembling CD82 alone was observed during particle cleaning in 3D in both data sets (Figs S3 and S4C and D). The processing of the second set of particles resulted in a density at 15 Å resolution (averaged over 13,848 particles). The density suggests EGFR curving back to the micelle consistent with our 2D classes and previous NMR and molecular dynamic studies that indicate interaction between EGFR(I) and the membrane (Arkhipov

et al, 2013; Kaplan et al, 2016) (Fig 3C). The achieved resolution was not sufficient to position EGFR or CD82 molecules reliably into the density.

We predicted the complex of EGFR (amino acids 1–1,009) and CD82 using AlphaFold2 multimer (Jumper et al, 2021; Mirdita et al, 2022) (Fig S6). AlphaFold models 1–3 showed non-physiological models with EGFR folding below the hypothetical cell membrane and interactions between transmembrane helices and intracellular domains. AlphaFold models 4 and 5 showed more sensible models with higher pTM scores of 0.424 and 0.418, respectively, and interactions between transmembrane helices and EGFR domains III and IV with CD82-LEL. The interaction between the β-hairpin of EGFR(IV) and a loop in CD82-LEL in AlphaFold models 4 and 5 is predicted with low local pLDDT values but is in agreement with our cryo-EM density (Figs S6 and 3C and D). In summary, our cryo-EM analysis of the EGFR(I-TM)-CD82 fusion protein in the presence of EGF-His₆ resulted in an EGF-bound EGFR dimer without CD82. This indicates that CD82 dissociates from EGFR after EGF binding. The resulting 3D volume in the absence of EGF indicated an interaction between monomeric EGFR domain IV and the LEL of CD82 at low resolution in agreement with AlphaFold2 predictions.

### TEV-cleavage assay of EGFRΔβ-hairpin-CD82 and EGFR-CD82 in the presence of EGF

To test the possible EGFR-CD82 interaction site between the β-hairpin of EGFR(IV) and CD82-LEL, we repeated the TEV-cleavage assay (Fig 2A–D) using EGFR(I-TM)Δβ-hairpin-CD82-eGFP-StrepII₃ in which amino acids 571–596 were deleted and EGFR(I-TM)-CD82-

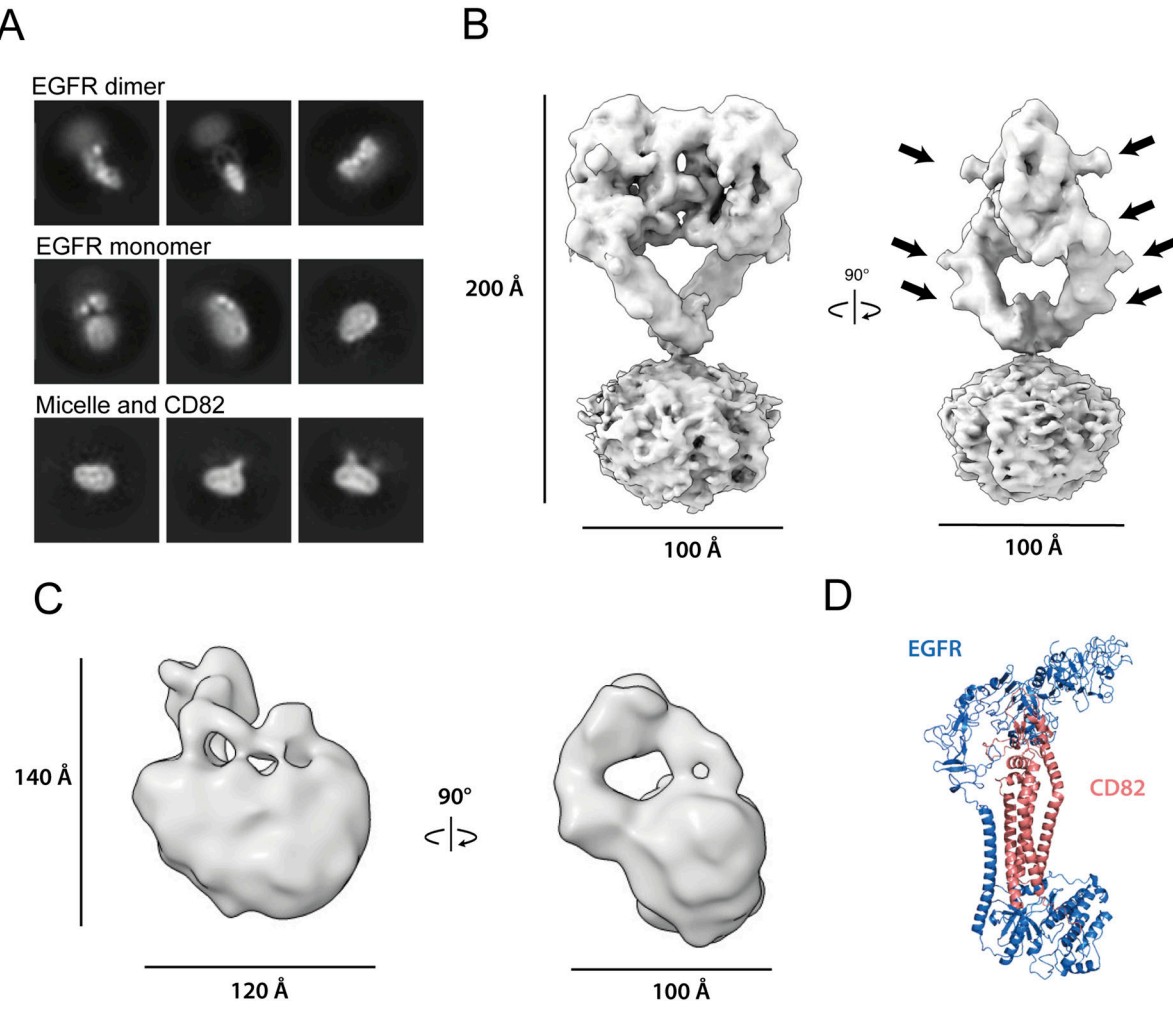

**Figure 3. Structure determination of the EGFR-CD82 complex.**
**(A)** 2D classes corresponding to dimeric EGFR, monomeric EGFR, and a micelle with two protein densities. All 2D classes were cropped around the mask. **(B)** Final 3D volume of EGF-bound EGFR dimers. N-glycans are marked with black arrows in the side view. **(C)** Final 3D volume of the EGFR-CD82 complex. **(D)** AlphaFold2 multimer prediction with the highest pTM score of EGFR(1–1,009) in blue and CD82 in salmon.
Source data are available for this figure.

eGFP-StrepII$_3$ in the presence of EGF-His$_6$. The SDS–PAGE gel showed a double band at 140 and 160 kD, similar to the observed double band of EGFR(I-TM)-CD82-eGFP-StrepII$_3$, and identity of the bands was confirmed by the eGFP signals (Fig 4A and B). After 2 h of cleavage with TEV protease on ice, nearly all EGFR(I-TM)Δβ-hairpin-CD82-eGFP-StrepII$_3$ was cleaved (Fig 4A and B). F-SEC of the Δβ-hairpin mutation showed a double peak at 10 and 10.5 ml that upon cleavage partially shifts to a double peak at 11 and 11.5 ml, therefore showing an additional peak for uncleaved and cleaved protein with 0.5 ml later EVs compared with the WT fusion protein (Fig 4C). Similar to WT, we observed initially fast dissociation that reduced over time. When adding EGF-His$_6$ to the EGFR(I-TM)-CD82-eGFP-StrepII$_3$, an additional peak appeared at 9 ml likely corresponding to dimers (Fig 4D). Apart from the additional peak, the F-SEC profile highly resembled the profile observed for WT fusion protein in the absence of EGF-His$_6$. To quantify and compare the F-SEC profiles, we integrated the areas under the curve (AUCs) of

co-eluting EGFR and CD82 and dissociated CD82 for the WT fusion protein, the Δβ-hairpin mutation, and the WT fusion protein in the presence of EGF-His$_6$ at each time point to calculate remaining co-elution. All showed an initially fast declining curve that flattens over time (Fig 4E). Although the addition of EGF-His$_6$ did not impact the dissociation curve of EGFR(I-TM)-CD82-eGFP-StrepII$_3$, the Δβ-hairpin mutation showed a faster dissociation rate. These observations are consistent with an interaction between the β-hairpin of EGFR and the CD82-LEL as the deletion of hairpin leads to faster dissociation and the binding site of EGF is not affecting the interaction before dissociation of the complex.

### Affinity measurements of EGF to the EGFR-CD82 fusion protein

We tested whether the fusion of CD82 to EGFR(I-TM) affects the EGF-binding affinity of the receptor. MicroScale Thermophoresis (MST) was applied using EGFR(I-TM)-CD82-StrepII$_3$ and EGFR(I-TM)-

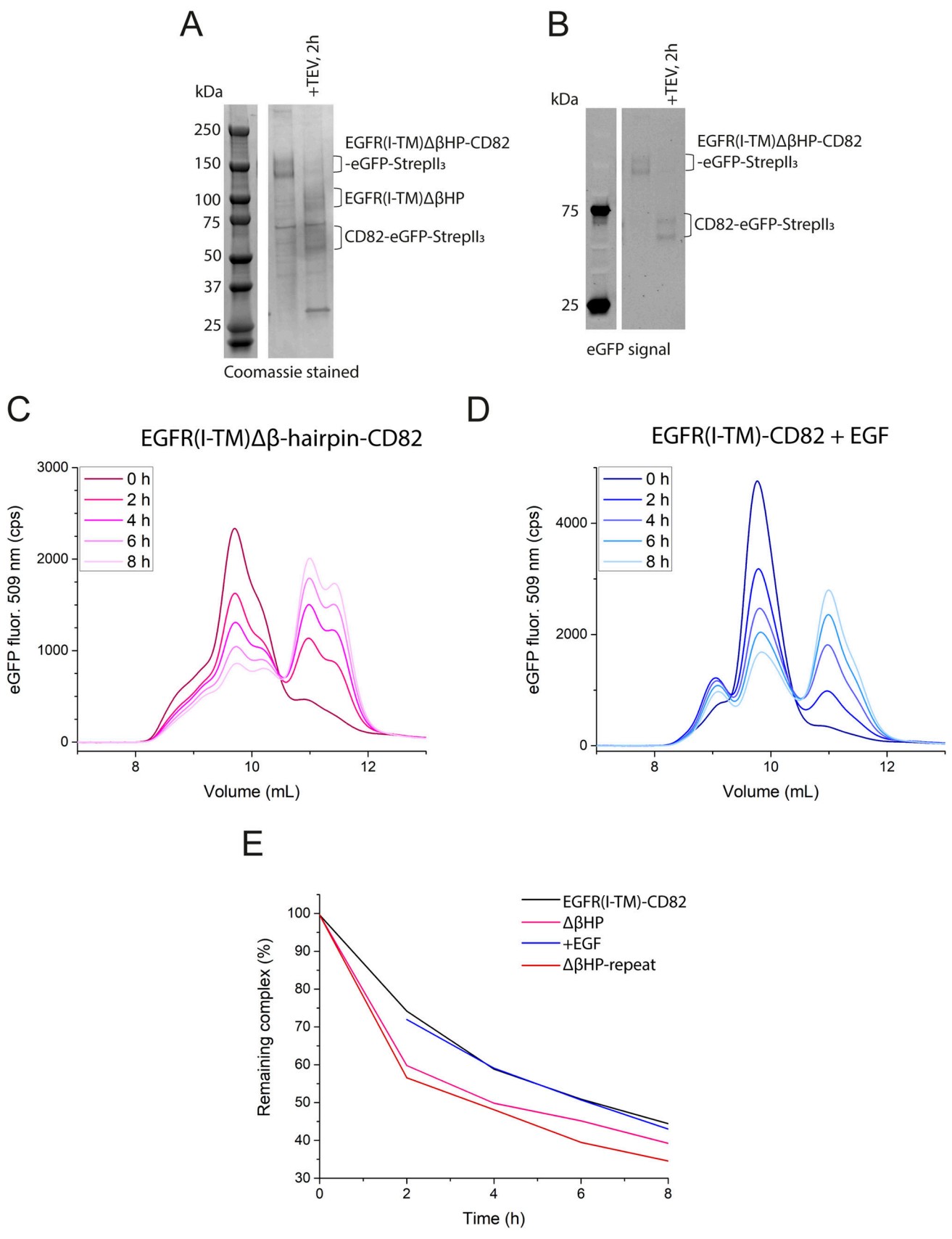

StrepII$_3$ (Fig S7A–D). His$_6$-EGF-eGFP was previously purified and labelled with DyLight 650 (Fig S8A and B). Variable concentrations of EGFR(I-TM)-StrepII$_3$ or EGFR(I-TM)-CD82-StrepII$_3$ were mixed with 5 nM DyLight 650–labelled His$_6$-EGF-eGFP. We did not observe a stable fluorescence signal for bound receptor at low concentrations for both tested proteins; instead, we observed a consistent decrease in fluorescence signal at receptor concentrations between 1.5 and 12 nM for EGFR(I-TM)-StrepII$_3$ and between 0.5 and 7.8 nM for EGFR(I-TM)-CD82-StrepII$_3$ (Fig S9A and B). This effect is likely caused by EGF-induced formation of EGFR dimers in which the dimerization partner has a higher affinity to EGF (Macdonald & Pike, 2008). Receptor dimerization also doubles the molecular weight of the complex influencing thermophoresis and possibly causing the observed change in fluorescence caused by EGF binding. We excluded data points measured at receptor concentrations of less than 15 nM, to ensure reliable K$_D$ calculations for monomeric EGFR (Fig S9A and B). Our MST measurements, using DyLight 650–labelled EGF with His$_6$- and eGFP-tag, resulted in an EGF affinity of 288 nM (68% confidence interval 172–482 nM) for detergent-solubilized EGFR(I-TM) (Fig 5A), which is comparable to 400 nM measured by surface plasma resonance of EGF and soluble EGFR domain III (Lemmon, 1997). Our MST measurements of EGF binding to EGFR, using the same His$_6$- and eGFP-tagged EGF, in the presence of CD82, EGFR(I-TM)-CD82, resulted in 360 nM (68% confidence interval 248–524 nM) (Fig 5B), which is not significantly different from the affinity measured in the absence of CD82. Thus, we did not observe an increase in EGF-binding affinity to detergent-solubilized EGFR(I-TM)-CD82.

## Impact of CD82 on EGFR dimerization and EGF dissociation

We performed F-SEC to determine the effect of CD82 on EGFR dimerization and complex stability with His$_6$-EGF-eGFP. EGFR(I-TM)-StrepII$_3$ and EGFR(I-TM)-CD82-StrepII$_3$ were purified and used fresh (Fig S10A). His$_6$-EGF-eGFP was incubated with EGFR or fusion protein at an EGF:receptor ratio of 2:1 before running an F-SEC that monitors tryptophan and eGFP fluorescence. The tryptophan fluorescence showed a broadened peak at 15 ml EV for EGFR(I-TM)-StrepII$_3$ (Fig 6A) and at 14.5 ml EV for EGFR(I-TM)-CD82-StrepII$_3$ (Fig 6B). His$_6$-EGF-eGFP induced dimerization of both proteins. EGFR(I-TM)-StrepII$_3$ dimers eluted at 13.5 ml EV and dimers of EGFR(I-TM)-CD82-StrepII$_3$ at 13 ml EV (Fig 6A and B). The peak of dimerized EGFR(I-TM)-StrepII$_3$ was about three times more intense compared with the peak of dimeric EGFR(I-TM)-CD82-StrepII$_3$. This indicates an inhibiting effect of CD82 on receptor dimerization, consistent with reports showing decreased receptor dimerization on CD82-expressing cells compared with CD82-KO cells (Odintsova et al, 2003). The eGFP fluorescence showed at which EV His$_6$-EGF-eGFP eluted. The unbound ligand eluted at

17.5 ml EV. Bound His$_6$-EGF-eGFP co-eluted mostly with dimeric proteins at 13 or 13.5 ml (Fig 6A and B). To quantify how much His$_6$-EGF-eGFP stayed bound, we integrated the AUCs of the eGFP curves of bound (12–16 ml EV) and free (16–19 ml EV) His$_6$-EGF-eGFP. His$_6$-EGF-eGFP dissociated ~4 times less from EGFR(I-TM)-StrepII$_3$ than from EGFR(I-TM)-CD82-StrepII$_3$ (Fig 6C). This difference in dissociation can be due to impaired EGF binding and to the inhibition of EGFR dimerization that decreases complex stability or both.

To assess the impact of EGFR dimerization on EGF-EGFR complex stability, we repeated the F-SEC experiments with an EGFR double mutant, Y251A and R285S, which are located in the domain II dimerization arm, preventing dimerization (Ogiso et al, 2002). Mutated EGFR(I-TM)-StrepII$_3$ and EGFR(I-TM)-CD82-StrepII$_3$ were purified (Fig S10B), and His$_6$-EGF-eGFP was incubated at a 1:1 ratio. The tryptophan fluorescence confirmed that EGFR alone and in fusion no longer dimerized (Fig 6D and E). The percentage of bound EGF decreased ~100 times for EGFR(I-TM)-StrepII$_3$ and EGFR(I-TM)-CD82-StrepII$_3$, supporting that EGFR dimerization is necessary to form a stable complex with EGF. There was no difference in bound EGF between EGFR(I-TM)-StrepII$_3$ and EGFR(I-TM)-CD82-StrepII$_3$ (Fig 6F), indicating that reduced EGFR dimerization by CD82 causes reduced EGF binding for EGFR(I-TM)-CD82-eGFP-StrepII$_3$. Both experiments were executed at different EGF:receptor ratios, which showed similar results (Fig S11A and B). In summary, our F-SEC studies showed that CD82 inhibits EGF-induced EGFR dimerization. CD82 negatively impacts EGFR-EGF complex stability of dimerization-capable EGFRs.

# Discussion

Previous studies showed that EGFR and CD82 interact and are colocalized in TEMs (Odintsova et al, 2000, 2003). Even though some biological consequences of this interaction are known, the underlying molecular mechanisms are unclear (Odintsova et al, 2000, 2003; Sugiyama et al, 2023). In this study, we purified the EGFR-CD82 complex from mammalian (HEK293-E+) cells via an affinity tag on CD82. EGFR and CD82 were copurified from co-expressions and co-eluted in a broadened, low-intensity peak in SEC with other unidentified proteins. When purifying a fusion protein of EGFR(I-TM) and CD82, we observed a higher and sharper peak in SEC, indicating higher yield and purity. Overall, we confirmed complex formation of EGFR and CD82 and could purify a fusion protein of the two at high purity.

SPA cryo-EM of the EGFR(I-TM)-CD82 fusion protein yielded a 15 Å resolution density of EGFR folding back onto the LEL domain of CD82. Similarly, AlphaFold2 predicted a folded-back model with interactions between the transmembrane helices of EGFR and

**Figure 4. TEV-cleavage assay of EGFR(I-TM)Δβ-hairpin-CD82 mutation and EGFR(I-TM)-CD82 in the presence of EGF.**
**(A, B)** 4–15% SDS–PAGE gel of EGFR(I-TM)Δβ-hairpin-CD82-StrepII$_3$ (selected from the same gel as in Fig 2B and C) before and after incubation with TEV protease for 2 h on ice (A) stained with Coomassie and (B) eGFP fluorescence. **(C)** F-SEC profile of EGFR(I-TM)Δβ-hairpin-CD82-StrepII$_3$ after incubation with TEV protease at different time points. **(D)** F-SEC profile of EGFR(I-TM)-CD82-StrepII$_3$ in the presence of EGF-His$_6$ after incubation with TEV protease at different time points. The curve at 0-h TEV cleavage was measured in the absence of EGF. **(E)** Quantification of the remaining complex over time by integrating the area under the curves before and after 10.5 ml elution volume. All F-SEC experiments were performed using a Superdex 200 Increase 10/300 Gl column.
Source data are available for this figure.

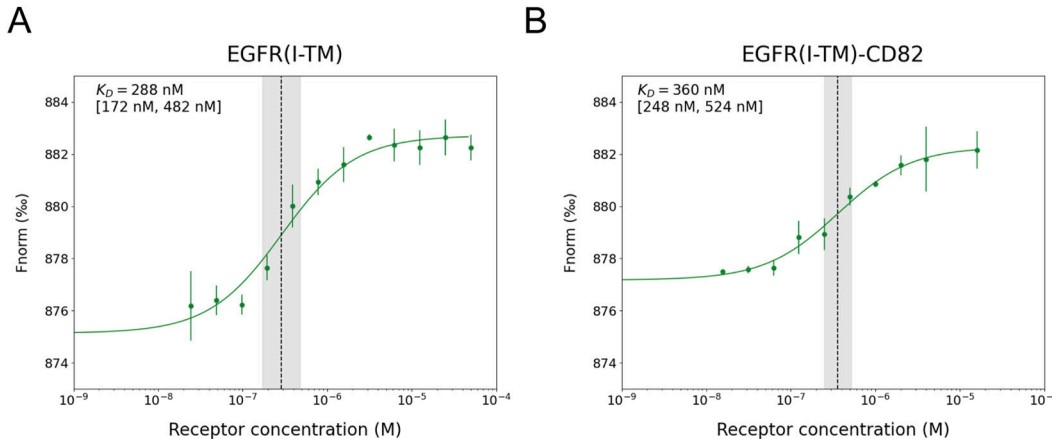

**Figure 5. MST measurements of red dye 650–labelled His$_6$-EGF-eGFP binding to EGFR(I-TM)-CD82-StrepII$_3$ and EGFR(I-TM)-StrepII$_3$.**
**(A, B)** 5 nM of red dye–labelled His$_6$-EGF-eGFP was mixed with a varying concentration of (A) EGFR(I-TM)-StrepII$_3$ (B) and EGFR(I-TM)-CD82-StrepII$_3$. The error of individual measurements is indicated as error bars. The resulting K$_D$ value is shown as a black, dotted line. The 68% confidence interval of the K$_D$ is indicated as a grey, translucent box and written in squared brackets.
Source data are available for this figure.

CD82 and an interaction of the β-hairpin of EGFR(IV) and CD82-LEL with low local pLDDT values. SPA cryo-EM of the fusion protein in the presence of EGF yielded EGF-bound EGFR dimers at higher resolution without discernible density for CD82. Furthermore, we observed double micelles in 2D classes and a 3D volume resembling CD82 alone. Thus, we hypothesize that CD82 dissociated from EGFR(I-TM) dimer–containing micelles into separated micelles, even though it remains fused to EGFR (I-TM). Consistently, single-particle tracking studies observed a reduction of the EGFR-CD82 colocalization after EGF binding, although Sugiyama et al argued that this was caused by reduced cluster size and not dissociation (Sugiyama et al, 2023). To test our hypothesis, we generated EGFR(I-TM)Δβ-hairpin-CD82 fusion protein and observed a faster complex dissociation rate after TEV cleavage of this Δβ-hairpin mutant compared with the EGFR(I-TM)-CD82 fusion protein. However, the observed dissociation rates of the detergent-solubilized membrane proteins are likely reduced because of the presence of micelles. Although our deletion experiment could confirm that the EGFR-β-hairpin has a stabilizing effect on the EGFR-CD82 complex, it does not exclude the presence of additional interaction sites. Overall, our data suggest CD82 interactions with monomeric EGFR through their ectodomains, putatively involving the β-hairpin of EGFR(IV) and CD82-LEL, and dissociation of CD82 upon EGF-induced EGFR dimerization.

To explore possible functional implication of the interaction in vitro, we analysed EGF binding to EGFR and EGFR dimerization in the presence and absence of CD82 using MST measurements and F-SEC. These showed no change in EGF-binding affinity but decreased EGF binding as a result of inhibited EGFR dimerization in the presence of CD82. Although this is in agreement with our TEV-cleavage assay and structural data, affinity measurements on an epithelial cell model showed EGF binding to EGFR is more pronounced in CD82-rich TEMs at short time intervals suggesting an effect on ligand association rate or possible further mechanisms on the cell level (Sugiyama et al, 2023). Our proposed model of an EGFR-CD82 interaction suggests CD82 binding via the β-hairpin on

domain IV of EGFR, which simultaneously blocks the formation of the closed conformation of EGFR and prevents dimerization. Blocking the closed conformation by CD82 binding would possibly result in a higher EGF association rate and consequent binding affinity: deletion of domain IV of soluble EGFR was shown to increase EGF-binding affinity (Elleman et al, 2001). However, several tested mutations that disrupt the closed conformation did not result in comparable binding affinities of the high-affinity state of EGFR (Mattoon et al, 2004; Özcan et al, 2006), whereas SAXS of mutated EGFR ectodomain did not show the adaptation of the extended conformation (Dawson et al, 2007). On the other hand, reduced receptor dimerization by CD82 is expected to increase EGF dissociation, as observed in our F-SEC experiments, and therefore stimulates dissociation rate and consequently reduces EGF-binding affinity (Özcan et al, 2006). Because receptor dimerization caused a biphasic binding curve in our MST measurements, we excluded measurements influenced by receptor dimerization, and thus, we only measured binding affinity to monomeric EGFR in the presence and absence of CD82. In conclusion, we demonstrated that CD82 destabilizes the EGFR-EGF complex indirectly by inhibiting receptor dimerization but did not detect a difference in EGF-binding affinity in the presence of CD82. This is consistent with our model of CD82 interacting with domain IV but might also be a result of CD82 sterically blocking close proximity of the transmembrane domains of EGFR, which is necessary for dimerization.

Early studies showed that EGF binding to high-affinity EGFRs and consequential signalling required an additional cellular factor (Defize et al, 1989; Berkers et al, 1990), consistent with the role of CD82. EGFR confinement in CD82-rich TEMs converts EGFR into a high-affinity state (Sugiyama et al, 2023), by blocking domain IV. Upon EGF binding, EGFR dimerizes leading to dissociation of CD82 potentially causing the observed decrease in TEM size (Sugiyama et al, 2023). After activation, signalling EGFR dimers were shown to assemble in cholesterol-rich, immobile clusters (Hofman et al, 2008; Mudumbi et al, 2024) that are internalized via clathrin-coated pits where signalling is terminated by degradation

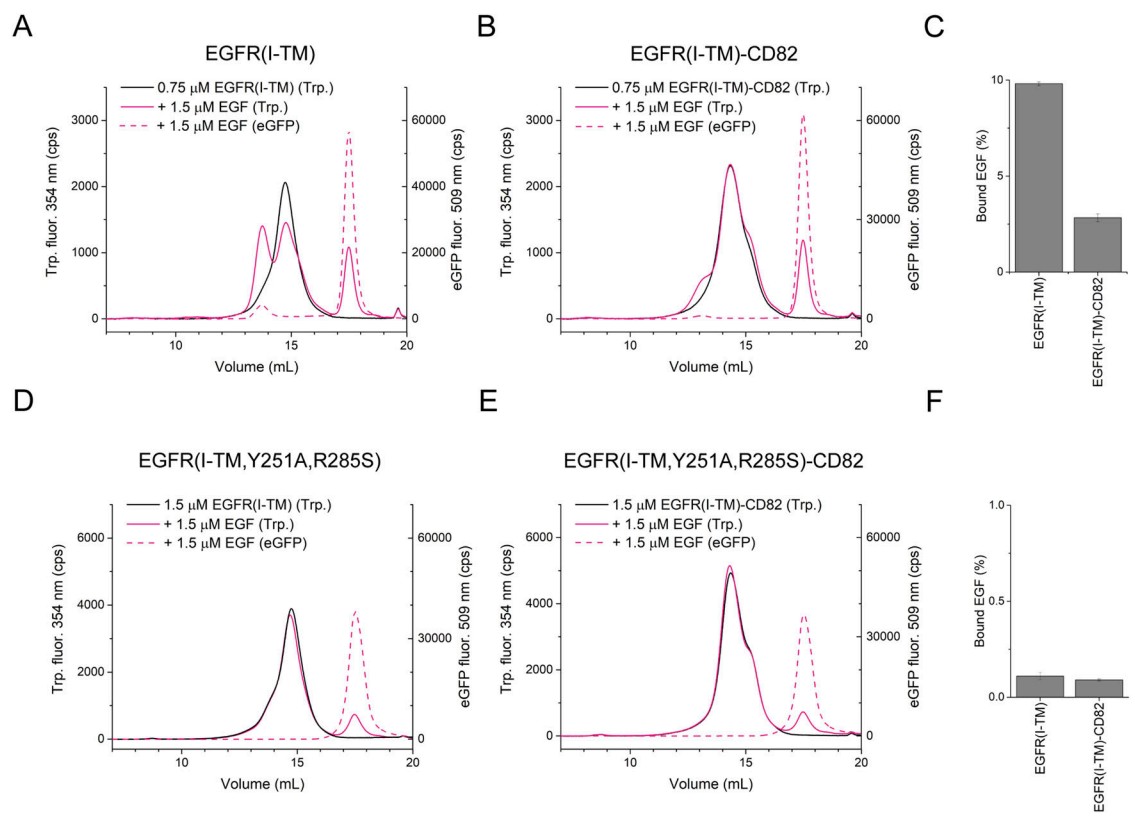

**Figure 6. EGF-induced dimerization and complex stability assay in the presence of CD82.**
**(A)** F-SEC profile showing Trp fluorescence and eGFP fluorescence of 0.75 μM EGFR(I-TM)-StrepII$_3$ alone and mixed with 1.5 μM His$_6$-EGF-eGFP. **(B)** F-SEC profile of 0.75 μM EGFR(I-TM)-CD82-StrepII$_3$ alone and mixed with 1.5 μM His$_6$-EGF-eGFP. **(C)** Percentage of bound EGF to WT proteins calculated from technical triplicates of the area under the curve from eGFP signals. **(D)** F-SEC profile of 1.5 μM EGFR(I-TM)-StrepII$_3$ alone and mixed with 1.5 μM His$_6$-EGF-eGFP. **(E)** F-SEC profile of 1.5 μM EGFR(I-TM)-CD82-StrepII$_3$ alone and mixed with 1.5 μM His$_6$-EGF-eGFP. **(F)** Percentage of bound EGF calculated from technical triplicates of the AUC of the eGFP signal. All F-SEC was executed using a Superose 6 Increase 10/300 Gl column.
Source data are available for this figure.

of EGFR (Heukers et al, 2013; Odintsova et al, 2013) (Fig 7). In this research, we studied detergent-solubilized EGFR(I-TM)-CD82 complexes and therefore did not consider more mechanisms than protein–protein interaction of the extracellular domains or transmembrane helices that play a role of EGFR regulation by CD82. A study about EGF binding to EGFR-expressing cells proposed an additional factor binding to the intracellular domain of EGFR (Özcan et al, 2006). This is supported by another study showing the absence of high-affinity EGF-binding kinetics for cell-derived, EGFR-containing membrane vesicles. This might be due to cellular factors, such as tetraspanins (Berkers et al, 1990). We used truncated EGFR(I-TM) and therefore did not consider possible intracellular factors nor possible effects of the intracellular domain of EGFR on EGFR dimerization and EGF-binding affinity as indicated by the analysis of tyrosine-kinase inhibitors (Macdonald-Obermann & Pike, 2018). The gangliosides G$_{D1A}$ and maybe G$_{M2}$ and G$_{M1}$ have a negative impact on EGFR confinement in CD82-rich TEMs (Odintsova et al, 2006; Park et al, 2009; Santos et al, 2022). At the same time, gangliosides interact directly with N-glycans on the EGFR (Miljan & Bremer, 2002; Kawashima et al, 2009). However, whether gangliosides influence EGF-binding affinity remains unclear (Hanai et al, 1988; Coskun et al, 2011). In a similar manner, another membrane lipid, cholesterol, was shown to induce a higher affinity of reconstituted EGFR and in addition a conformational change to tetraspanin CD81 that might inhibit CD19 binding (den Hartigh et al, 1993; Susa et al, 2021). Future experiments can focus on the influence of CD82 on EGFR function in a controlled lipid environment.

EGFR can be activated by six other high- and low-affinity ligands, such as TGF-α and epiregulin, and coprecipitates with two other tetraspanins, CD9 and CD81 (Murayama et al, 2008; Park et al, 2009). The interaction of EGFR with different ligands has been shown to effect dimerization of the tyrosine-kinase domain, which leads to differences in phosphorylation patterns of the C-terminal tail and activation of different signalling cascades (Macdonald-Obermann & Pike, 2014; Doerner et al, 2015). In a similar way, the interaction of CD9 with the EGFR attenuates its signalling (Murayama et al, 2008). Further experiments could focus on the combined effect of different tetraspanins and ligands on EGFR mechanics and signalling. Possibly, the interacting tetraspanin has an effect on ligand specificity of EGFR in a comparable manner as substrate specificity of ADAM10 is impacted by the interacting tetraspanin (Jouannet

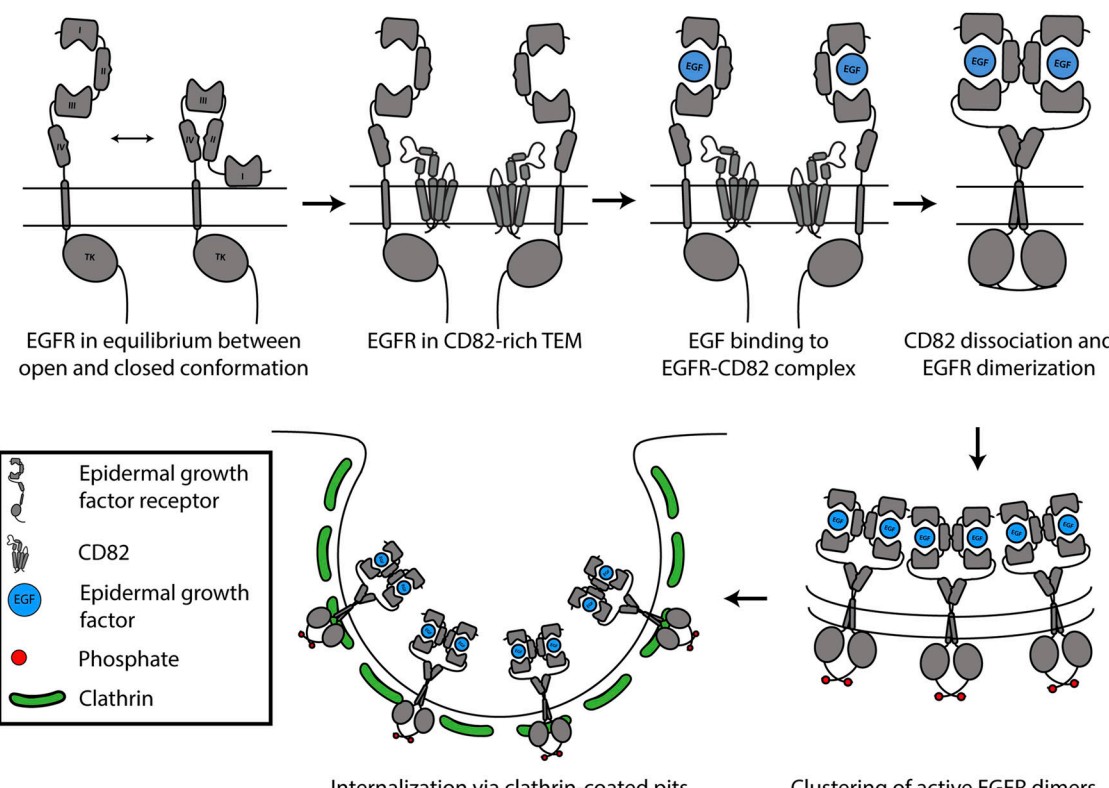

**Figure 7. Overall mechanism of EGFR regulation by CD82.**
The proposed model of EGFR activation and degradation is impacted by CD82. EGFR and CD82 are shown in grey, and EGF in blue; phosphorylation of EGFR is indicated by the addition of a red sphere; and the formation of clathrin-coated pits is indicated by clathrin models in green as a starting point of EGFR internalization.

et al, 2015; Lipper et al, 2023). There might be a comparable mechanism for EGFR. The here presented possible mechanism on EGFR regulation by CD82 is likely to be supplemented by more cellular control mechanisms.

# Materials and Methods

### Constructs

The DNA encoding EGF (UniProtKB: P01133), EGFR (UniProtKB: P00533-1), CD82 (UniProtKB: P27701-1, VAR_052326), and the linker used for fusion proteins (linker: AA-4xGGS-ENLYFQG-4xGGS-RS) was ordered from GeneArt. DNA coding for full-length EGFR was ligated into expression vectors containing a signal peptide and a C-terminal His$_6$-tag. Full-length CD82 was cloned into an expression vector containing a C-terminal StrepII$_3$- or eGFP-StrepII$_3$-tag. DNA coding for EGFR was truncated after the transmembrane helix (1–672) and ligated into an expression vector containing an N-terminal signal peptide and C-terminal StrepII$_3$-tag. The same EGFR construct was linked to full-length CD82 via the ordered linker, and DNA coding for the fused protein EGFR(I-TM)-CD82 was cloned into the same expression vector as EGFR(I-TM). The Y251A and R285S double mutation of EGFR(I-TM) and EGFR(I-TM)-

CD82 was included by mutagenesis PCR. DNA coding for EGF was ligated into an expression vector containing an N-terminal signal peptide and a C-terminal His$_6$-tag and an expression vector containing an N-terminal signal peptide and a His$_6$-tag and C-terminal eGFP-tag. All expression vectors were provided by ImmunoPrecise Antibodies (Europe) BV. All enzymes used for molecular cloning were purchased from New England Biolabs.

### Small-scale expression and purification of membrane proteins

DNA coding for membrane proteins was transiently transfected into 4 ml or 25 ml of HEK293-E+ cells (ImmunoPrecise Antibodies [Europe] BV) by polyethylenimine treatment. Cells were harvested 4 d post-transfection by centrifugation (1,000$g$, 10 min, 4°C). Cell pellets were used fresh or flash-frozen in liquid nitrogen and stored at –80°C. All the following purification steps were carried out at 4°C or on ice and with pre-cooled buffers. Cell membranes were solubilized with lysis buffer containing 150 mM NaCl, 25 mM Hepes, pH 7.5, 1% wt/vol n-dodecyl-$\beta$-D-maltopyranoside (DDM), and EDTA-free cOmplete protease inhibitor tablets (Roche) with gentle mixing for 1 h. Cell lysates were centrifuged (100,000$g$, 45 min), and the supernatant was incubated with Strep-Tactin Sepharose beads (GE Healthcare) for 2 h with gentle mixing. Beads were washed three times with 10 column volume (CV) washing buffer containing 150 mM NaCl, 25 mM Hepes, pH 7.5, and

0.025% wt/vol DDM. Protein was eluted using washing buffer supplemented with 3.5 mM D-desthiobiotin. Anti-His Western blots were treated with a mixture of Monoclonal Anti-polyHistidine, Clone His-1 (Sigma-Aldrich) and Penta-His Antibody, Mouse anti-(H)5 (QIAGEN) diluted 1:2,000 in blocking buffer (3% BSA in PBS + 0.05% Tween-20) as primary antibodies and 1:10,000 Goat anti-mouse IgG, HRP (Bio-Rad) as a secondary antibody.

### Large-scale expression and purification of membrane proteins

For large-scale expression in 200 ml, 1 litre, or 2 litre HEK293-E+ cells, protein expression was carried out as described for small-scale expressions. All steps were carried out at 4°C or on ice and with pre-cooled buffers. The cell pellet was washed once with PBS. Cell pellets were processed fresh or flash-frozen in liquid nitrogen and stored at −80°C. The cells were solubilized using lysis buffer containing 150 mM NaCl, 25 mM Hepes, pH 7.5, 10% vol/vol glycerol, 10 $\mu$g/ml DNase (Sigma-Aldrich), 1% wt/vol DDM, 0.2% wt/vol cholesteryl hemisuccinate (CHS), and EDTA-free cOmplete protease inhibitor tablets (Roche) with gentle mixing for 2 h. CHS was not used in the first large-scale purification (Fig 1D and E). The lysate was centrifuged (100,000$g$, 45 min), and the supernatant was incubated with Strep-Tactin Sepharose beads (GE Healthcare) for 2 h with gentle mixing. Beads were washed with washing buffer, containing 150 mM NaCl, 25 mM Hepes, pH 7.5, 0.025% wt/vol DDM, and 0.005% wt/vol CHS. The protein was eluted with 3.5 mM d-desthiobiotin in washing buffer. For all experiments except the EGFR dimerization and EGF-EGFR complex stability assay, the protein was further purified by size-exclusion chromatography (SEC). The protein was injected onto a Superdex 200 Increase 10/300 GL or Superose 6 Increase 10/300 GL column (GE Healthcare Life Sciences) equilibrated with washing buffer. Fractions with protein of interest were pooled and concentrated using a 100 kD MWCO filter (Amicon).

### Expression and purification of EGF

Protein expression was carried out as described for membrane proteins. The DNA was transfected into 400 ml for His$_6$-EGF-eGFP expression and 1 litre HEK293-E+ cells for EGF-His$_6$ expression, and cells were harvested 6 d post-transfection. The cells were pelleted twice by centrifugation (5,000$g$, 15 min, RT) and discarded. The supernatant was circulated over a 5-ml prepacked HisTrap Excel column (Cytiva) at 4°C overnight. The column was washed with 30 CV buffer, containing 500 mM NaCl, 25 mM Hepes, pH 7.5, and 25 mM imidazole, and protein was eluted using buffer with 500 mM NaCl, 25 mM Hepes, pH 7.5, and 500 mM imidazole. The proteins were further purified by SEC, using a Superdex 75 Increase 10/300 column or a HiLoad 16/600 75 pg column (GE Healthcare Life Sciences) equilibrated with running buffer (200 mM NaCl, 25 mM Hepes, pH 7.5). Fractions containing protein were pooled and concentrated using 5 kD MWCO or 10 kD MWCO filters (Amicon). Proteins were aliquoted, flash-frozen in liquid nitrogen, and stored at −80°C until further use.

### EGFR(I-TM)-CD82 complex formation after TEV cleavage

The fusion protein of EGFR(I-TM)-CD82-eGFP-StrepII$_3$ was expressed in a 25 ml HEK293-E+ culture and purified as described. 2 $\mu$g EGFR(I-TM)-CD82-eGFP-StrepII$_3$ was incubated with 10:1 (OD280) TEV protease between 0 and 8 h in 2-h intervals at 4°C or on ice. After incubation, samples were injected onto a Superdex 200 Increase 10/300 GL (GE Healthcare Life Sciences) connected to a Prominence UFLC system (Shimadzu) and running buffer containing 150 mM NaCl, 25 mM Hepes, pH 7.5, and 0.025% wt/vol DDM. The fluorescence signal of the eGFP (emission: 488 nm; excitation: 509 nm) was detected by a RF-10AXL fluorescence detector (Shimadzu). The amount of EGFR(I-TM)-CD82 complex was determined by integrating the peak at 10 ml EV using Origin 9.1.

### Grid preparation and cryo-EM data collection for SPA

EGFR(I-TM)-CD82-StrepII$_3$ was expressed in 2 × 1 litre HEK293-E+ cells and purified as described. We added 2 × molar excess of EGF-His$_6$ to one of the purifications after the affinity purification step. 3.5 $\mu$l of protein at 3 or 5 mg/ml was pipetted on a glow-discharged R1.2/1.3 300 mesh Au holey carbon grid (Quantifoil). Grids were plunge-frozen in liquid ethane using a Vitrobot Mark IV system (Thermo Fisher Scientific) for 3.5 s at 8°C and 90% humidity. Grids were clipped and stored in liquid nitrogen until imaging. Movies were collected using a 200 kV Talos Arctica microscope (Thermo Fisher Scientific) equipped with a K2 direct-detection camera (Gatan) using AFIS in SerialEM at a magnification of 100,000 resulting in a pixel size of 1.36 Å/px with a defocus range of −2.5 to −1 $\mu$m. For the data set with EGFR(I-TM)-CD82 fusion protein, 3,133 movies were collected with a total dose of 50 e−/Å$^2$ distributed over 35 frames. For the data set of EGFR(I-TM)-CD82 with EGF, 2,819 movies were collected with a total dose of 57 e−/Å$^2$ distributed over 40 frames.

### Cryo-EM data processing

Data processing was carried out in CryoSPARC v3.3. Final refinements were made in CryoSPARC v4.5.3. Patch-based motion correction and CTF estimation were performed followed by manual curation of exposures. Post-curation, 2,921 movies of the EGFR(I-TM)-CD82 data set and 2,784 movies of the data set with EGF were used for further processing. For 2D classification of both data sets, blob picking was used followed by several rounds of template picking. After 2D classification, particles were separated into two classes. Particles in 2D classes representing dimerized EGFR and particles in 2D classes representing EGFR-CD82 complex were processed separately. Particles were cleaned in 3D using several rounds of Ab initio models and heterogeneous refinement for both data sets. For the EGFR-CD82 data set, particles of a promising 3D model were used for training and picking new particles using TOPAZ (Bepler et al, 2019). 2D classification and particle cleaning in 3D were carried out as previously described. Masks used for local refinements were created using the Map-eraser tool in ChimeraX v1.8 (UCSF). An overview of the data processing pipeline is illustrated in Fig S3. A previously solved structure of EGF-bound EGFR

dimer (PDB: 7SYD) was fitted into the density using UCSF ChimeraX 1.8.

## MicroScale Thermophoresis

$His_6$-EGF-eGFP was labelled with DyLight 650 (Thermo Fisher Scientific), and labelled protein was flash-frozen in liquid nitrogen and stored at –80°C until use. EGFR(I-TM)-StrepII$_3$ and EGFR(I-TM)-CD82-StrepII$_3$ were both expressed in 1 litre HEK293-E+ cells and purified as described previously. Measurements were performed in washing buffer for membrane protein purifications containing 0.025% wt/vol DDM and 0.005% wt/vol CHS using a Monolith (NanoTemper Technologies). 5 nM of labelled $His_6$-EGF-eGFP was mixed with a 16 times 1:1 dilution series of EGFR(I-TM)-StrepII$_3$ at concentrations between 50 μM and 1.5 nM and with 16 times 1:1 dilution series of EGFR(I-TM)-CD82-StrepII$_3$ at concentration between 16 μM and 0.5 nM. All proteins were centrifuged at 20,000$g$, 4°C, for 5 min before being loaded into silica capillaries (NanoTemper Technologies). The MST measurements were carried out at 25°C, medium MST power, and 60% excitation power. Measurements were performed in technical triplicates for EGFR(I-TM)-CD82-StrepII$_3$ and technical quadruplicates for EGFR(I-TM)-StrepII$_3$ except for the 50 μM concentration. Extreme outliers and low receptor concentrations displaying negative cooperativity (Macdonald & Pike, 2008) were excluded from the graphs (Fig S9A and B).

## EGF dissociation and EGFR dimerization assay

EGFR(I-TM)-CD82-StrepII$_3$, EGFR(I-TM)-StrepII$_3$, EGFR(I-TM, Y251A, R285S)-CD82-StrepII$_3$, and EGFR(I-TM, Y251A, R285S)-StrepII$_3$ were expressed in 200 ml HEK293-E+ cells and purified as described previously. For this experiment, no SEC was performed for protein purification. Purified EGFR(I-TM)-CD82-StrepII$_3$ and EGFR(I-TM)-StrepII$_3$ were incubated with 1.5 μM $His_6$-EGF-eGFP at the molar EGF:receptor ratios of 2:1, 1:1, and 1:2 for 5 h in total with gentle mixing. The samples were centrifuged at 20,000$g$, 4°C, for 5 min before splitting in three samples for F-SEC. The F-SEC was performed using a Superdex 200 Increase 10/300 GL column (GE Healthcare Life Sciences) connected to a Prominence UFLC system (Shimadzu) and running buffer containing 150 mM NaCl, 25 mM Hepes, pH 7.5, 0.025% wt/vol DDM, and 0.005% wt/vol CHS. The fluorescence of tryptophan (emission: 275 nm; excitation: 354 nm) and eGFP (emission: 488 nm; excitation: 509 nm) was detected by a RF-10AXL fluorescence detector (Shimadzu). The percentage of bound $His_6$-EGF-eGFP was determined by integrating the corresponding peaks for bound or unbound $His_6$-EGF-eGFP in the eGFP signal using Origin 9.1 (OriginLab Corp.).

## Data visualization

All graphs showing SEC and F-SEC profiles and visualizations of peak integrations were created in Origin 9.1 (OriginLab Corp.). Binding affinity curves were created using PyCharm Community Edition 2024.2 (JetBrains). All figures containing cryo-electron microscopy densities were created using ChimeraX 1.8 (UCSF). Final figures were prepared using Illustrator (Adobe).

## Data Availability

Electron density maps of the EGF-bound EGFR(I-TM) dimer have been deposited in the EMDB with the ID EMD-53933, and electron density maps of the EGFR(I-TM)-CD82 complex have been deposited with the ID EMD-53883.

## Supplementary Information

## Acknowledgements

We thank ImmunoPrecise Antibodies (Europe) BV for transfections and protein expression, the Protein Research Centre (Utrecht University) for providing the instrument for MicroScale Thermophoresis measurements, R Koopman for helpful feedback on optimizing MST measurements, the Electron Microscope Centre (Utrecht University) for providing all necessary equipment for electron microscopy data collections, JS Depelteau, M Bergmeier, and SC Howes for their support during collections, and M Vanevic for computational support during processing. We thank P van Bergen en Henegouwen and D El Mazouni for insightful suggestions on experiments and P van Bergen en Henegouwen and FI Jäger for discussions and constructive feedback on the article. P Gros is supported by the Dutch Research Council (NWO) (grant number: 024.002.009).

### Author Contributions

E Lamottke: conceptualization, formal analysis, validation, investigation, visualization, methodology, and writing—original draft.
THC Brondijk: conceptualization, supervision, validation, methodology, project administration, and writing—review and editing.
P Gros: conceptualization, supervision, funding acquisition, validation, project administration, and writing—review and editing.

### Conflict of Interest Statement

The authors declare that they have no conflict of interest.

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
