## [Reviewer comments · Life Science Alliance]

BIOCHEMICAL AND STRUCTURAL DATA INDICATE MECHANISM OF EGFR REGULATION BY TETRASPANIN CD82

Elisa Lamottke, T. Harma Brondijk, and Piet Gros
DOI: <https://doi.org/10.26508/lsa.202503426>

Corresponding author(s): Piet Gros, Utrecht University

Review Timeline:

Submission Date:	2025-06-19
Editorial Decision:	2025-07-28
Revision Received:	2026-01-27
Editorial Decision:	2026-02-20
Revision Received:	2026-03-20
Editorial Decision:	2026-03-31
Revision Received:	2026-04-17
Accepted:	2026-04-28

Scientific Editor: Sarita Hebbar

Transaction Report:

July 28, 2025

Re: Life Science Alliance manuscript #LSA-2025-03426-T

Prof. Piet Gros
Utrecht University
Dept. of Chemistry
Crystal and Structural Chemistry
Utrecht 3584 CH
Netherlands

Dear Dr. Gros,

Thank you for submitting your manuscript entitled "Mechanistic implications on EGFR regulation by tetraspanin CD82" to Life Science Alliance. The manuscript was assessed by three expert reviewers, whose comments are appended to this letter.

Although the reviewers express potential interest in this work, significant concerns unfortunately preclude publication of the current version of the manuscript. The reviewers were consistent in pointing out that the cryoEM data is not of sufficient quality. In this regard, we tend to agree with Reviewer 1 and 3 that that you must either provide data from mutant studies OR improve the cryo-EM data to be considered for publication. We therefore invite you to submit a substantially revised version of the manuscript with additional experimental data to satisfactorily address the concerns raised in the evaluation.

The typical timeframe for revisions is three to four months. If you choose to revise and resubmit your manuscript, please include a letter addressing the reviewers' comments point by point. While a rebuttal must respond to all points in some form, additional experiments to resolve these points other than indicated above, is not required.

Thank you for this interesting contribution to Life Science Alliance. We are looking forward to receiving your revised manuscript.

Sincerely,

Sarita Hebbar, PhD
Scientific Editor
Life Science Alliance
<http://www.lsa-journal.org>

-- Summary blurb (enter in submission system): A short text summarizing in a single sentence the study (max. 200 characters including spaces). This text is used in conjunction with the titles of papers, hence should be informative and complementary to

the title and running title. It should describe the context and significance of the findings for a general readership; it should be written in the present tense and refer to the work in the third person. Author names should not be mentioned.

B. MANUSCRIPT ORGANIZATION AND FORMATTING:

Reviewer #1 (Comments to the Authors (Required)):

The Epidermal Growth Factor Receptor (EGFR) is a central driver of cellular growth and proliferation. Aberrant EGFR expression and/or activation drives tumor progression in several types of cancer. The understanding of EGFR regulation, including cellular factors that regulate receptor ligand binding, dimerization, and signal activation remain incomplete. Previous studies have implicated tetraspanins, including CD82 in regulation of EGFR membrane traffic and signaling. This study uses biochemical and structural approaches to examine the interaction of the tetraspanin CD82 with EGFR, and the functional outcome of this interaction on EGFR ligand binding and dimerization.

To do so, the extracellular and transmembrane domains of EGFR were expressed either alone or fused to CD82, the latter separated by a linker and TEV cleavage site. Each construct also contained eGFP and a StrepII3 tag for purification. Following expression and purification of proteins from HEK293 cells, the cleavage of EGFR from CD82 was confirmed by a range of methods including gel electrophoresis and fluorescent size exclusion chromatography (F-SEC). Notably, a significant fraction of EGFR and CD82 co-eluted in F-SEC, suggesting that interactions are retained following cleavage. The purified proteins were examined with SPA cryo-EM to resolve the structure of an EGFR-CD82 complex, either alone or in the presence of EGF-His6. This analysis appears to show several subpopulations of structures, one of which appears to show EGFR in complex with CD82 at 11-15 Å resolution. Fitting of the structures of EGFR and CD82 within this density suggests interaction between domain IV of the EGFR ectodomain and CD82. This interaction was consistent with the predicted interaction by AlphaFold2 using full-length EGFR and CD82. Microscale thermophoresis indicated that CD82 did not significantly impact the affinity of EGF for the EGFR ectodomain. F-SEC revealed that CD82 may disfavor EGF-stimulated EGFR dimerization, which is consistent with the observations that a Y251A and R285S mutant of EGFR that has impaired EGF-induced dimerization did not exhibit changes association of EGFR with EGF in the presence CD82.

This study proposes an intriguing model of CD82 interaction with domain IV of the EGFR ectodomain. This region of EGFR has been previously shown to be involved in regulation of EGFR tethering into a closed conformation with reduced ligand binding. This work suggests that CD82 interaction with EGFR may not impact the affinity of EGF with EGFR, but could regulate EGFR dimerization. This is a novel model that is reasonably well supported by the data presented. The study uses a range of methods including structural determination and assays to measure complex formation and structural changes. The co-expression of the EGFR ectodomain as a cleavable fusion protein with CD82 is a very useful strategy, ensuring similar levels of expression of both proteins within a sample to support subsequent assays. The expression of proteins in HEK293 cells is expected to ensure suitable glycosylation, an important consideration for EGFR and CD82. The manuscript also includes a thorough and careful consideration of the relevant literature, allowing the results of the study to be understood in the context of the leading edge of the field of EGFR signaling. Some comments below can be considered. This study will be of interest to a range of cell and cancer biology researchers.

Major comments:

1. The cryo-EM structure of EGFR with CD82 is somewhat limited by its resolution of 11-15 Å, which does not appear to allow unambiguous determination of proteins within the observed densities (at least not the TM domains, as stated in the manuscript). The prediction of CD82 interaction with domain IV of the EGFR ectodomain is an important and major finding of this study. It would this be very informative to consider experiments with either deletion of the EGFR ectodomain domain IV, or point mutation of key residues involved in this interaction with CD82. For instance, an experiment examining complex formation between the EGFR ectodomain and CD82 as in Figure 2 (e.g. by F-SEC), showing that this interaction is lost in a domain IV mutant of EGFR

would support this structural model.

Experiments as suggested with this mutant could also provide additional important controls for the interpretation of the F-SEC elution as being able to detect specific CD82-EGFR complex formation. As stated in the description of Figure 2 (page 4, line 136-137), the fact that these proteins are embedded in a common micelle may complicate the observed dissociation of EGFR and CD82, as this dissociation would be expected to be influenced not only by the protein-protein interaction but also by the confinement of these proteins within a common micelle. As such, it is not clear that the F-SEC co-elution data can unambiguously support a complex formed between EGFR and CD82.

This suggestion of additional experiments with an EGFR domain IV ectodomain mutant may represent a significant amount of work. The authors may consider tempering some conclusions instead, but experiments with such an EGFR mutant would significantly strengthen some key conclusions and the structural model.

2. The strategy taken to express and purify an EGFR truncation comprised of the ectodomain and the transmembrane domain instead of the full-length EGFR may be necessary from a practical perspective. However, the kinase domain influences dimerization and also impacts ligand-binding affinity (e.g. PMID: 29997256). This should be discussed as a potential limitation of this study.

3. The EGFR residues mutated to study the impact of EGF and CD82 binding (Y251A and R285S) appear to be in domain II, not domain IV of the ectodomain (as stated in lines 252-254 on page 8). An additional suggested reference for this could be PMID: 14732693.

4. How does the addition of a His6 tag to EGF impact its interaction with EGFR? It would be helpful to consider an assay to measure whether this tag has a significant impact of the interaction, such as perhaps comparing the ability of this EGFR-His6 to stimulate EGFR phosphorylation in cells compared to recombinant, commercially available EGF.

Minor comments:

1. The reference to the open, extended conformation of the EGFR ectodomain as the "active conformation" may be confusing, as EGFR activation typically refers to the formation of dimers with asymmetric arrangements of the kinase domains. Perhaps an alternative nomenclature can be used for the open, extended conformation of the EGFR ectodomain.

2. The structure of EGFR and CD82 shown in Figure 3 is a little difficult to see. The colors of some of the EGFR ectodomain regions (especially orange) are difficult to resolve from the red of the CD82. It would also be helpful to label each of these domains right on the image, rather than relying on the reader to match colors with statements in the figure caption.

Reviewer #2 (Comments to the Authors (Required)):

Lamottke et al. report biochemical and structural studies characterizing the interaction between EGFR and the tetraspanin CD82. I looked forward to reading this paper as, although many tetraspanins including CD82 have been reported to modulate EGFR activity, exactly how tetraspanins influence EGFR activity remains mysterious. I'm afraid, however, that this MS does little to reduce this mystery, and I cannot recommend publication. In particular, the cryo-EM studies of the EGFR-CD82 "complex" involve unconvincing fits of distorted EGFR structures into what appear noisy low-resolution maps.

Specific comments:

Line 46 "...the active conformation can form unstable dimers in the absence of ligand." Is that what is shown? Oligomers are present in the absence of ligand but I believe nothing about the nature or conformation of the oligomer in the absence of ligand is known.

Line 50 "In these self-activating oligomers, EGF-bound EGFRs flank a chain of unbound EGFRs (Liu et al. 2012)." I am puzzled by this statement, which is not reported or shown the cited reference (or elsewhere that I am aware of).

Line 51, I don't believe Needham et al. (2016) show direct evidence for "face-to-face" or "back-to-back" dimers. They simply show evidence for EGFR oligomers and make models. A more recent study of EGFR dimer clusters (Gonzalez-Magaldi et al. PNAS 2025) shows no regular interactions between EGFR dimer extracellular regions in EGFR clusters in vesicles studied by tomography.

Line 64 "The structure of a CD19-CD81 fusion protein reveals interactions between the extracellular domains (Susa et al. 2020)." Susa et al. model interactions between CD19 and CD81 but no complex structure is determined.

Line 105 "likely represent two different glycosylation species" What is MW of CD82 and how many glycosylation sites does it have?

Does Figure 1E really show co-migration? Does a Western blot confirm that EGFR-His6 is present in earlier elution fractions?

It would be helpful to label figures 2B and 2C Coomassie and GFP, respectively.

Line 349 what is the transfection method/reagent?

Line 179 "This suggests that CD82 dissociated from EGFR (I-TM), even though they were still fused." What does this mean? Was the fusion protein cleaved? If so, is there evidence for the cleavage? If not, why is low-res density for CD82 not seen?

Lines 150. It is not stated, but I presume EGF was only added to one of the two samples prepared?

Line 163 "The 2D classification of both samples resulted in three subgroups of particles. The first subgroup...was only observed for [the] sample containing EGF-His6". These statements are confusing. Were 3 different subgroups found for + vs. - EGF? Were similar subgroups found? It would help to be clear about which groups were found in + vs. - EGF samples.

Line 185 "Individual domains of the closed conformation of EGFR...were fitted into the density." From Figure 3 it looks like the individual domains were modeled independently despite there being no evidence that the domains move independently in the absence of ligand and that even in the presence of ligand most interdomain movement occurs between domains II and III. The fits also look wholly unpersuasive-given the poor quality of the maps and the contortions of the EGFR domains, I see no reason at all to believe the model.

Line 253 "...we repeated the F-SEC experiments with an EGFR double mutant, Y251 and R285S, which are located in the domain IV dimerization arm...". These amino acids are in the domain II dimerization arm.

Line 316 "EGFR can be activated by six other high and low affinity, such neuregulin and TGFalpha...". Neuregulin is not among the other six ligands that bind to EGFR.

Reviewer #3 (Comments to the Authors (Required)):

Summary:

Lamottke et al. present a very nice story on the interaction between EGFR and CD82. I really enjoyed reading this paper and found it to be well written and interesting. They describe protein engineering efforts to isolate a stable EGFR-CD82 complex, present a low-resolution structure of EGFR bound to CD82, and provide biochemical evidence that CD82 inhibits EGFR dimerization and promotes EGF dissociation.

Comments:

1. EGFR-CD82 refinement only had about 15k particles- is it possible to collect more data to improve the quality of the map? This would greatly improve the story. Are there any EGFR antibodies that you could use as fiducials to try to improve the map?
2. Line 178: "This suggests that CD82 dissociated from EGFR(I-TM), even though they were still fused" How do you think this is happening? Is the linker still present in your sample, or did you add TEV protease? If the linker was still present, are there any classes with two micelles close together (one with EGFR/EGF, and one with CD82?) Please update the text to clarify- It's not clear to me how CD82 would be dissociating if it is fused
3. Add a final model figure describing your model in lines 266-283
4. It is very interesting that CD82 inhibits EGFR dimerization. Is there any cell based evidence that overexpressing CD82 in cells attenuates EGFR signaling?
5. Line 64: The CD19-CD81 structure also showed interactions between the TMs, and it seems (from what we know now) that most Tspans interact with their partners through both the extracellular and TM region- please update text to clarify. The reference for this should also be Susa et al, 2021 (Science paper, not the eLife paper).
6. Line 67: There are 6 members of the TspanC8 family shown to interact with ADAM10, not 10.
7. Figure 2D: Using colors instead of grey scale would make this figure easier to quickly interpret for readers.
8. Line 149-150: Update to say "Two samples of EGFR(I-TM)- CD82-StreplI3 fusion protein were prepared, one with EGR, and one without."

9. Figure 3B: Please color EGF and EGFR different colors and add labels

10. Figure 3C: Please add labels for EGFR and the Tspan

11. Line 322: "Further experiments could focus on the combined effect of different tetraspanins and ligands on EGFR mechanics and signalling. A subgroup of tetraspanins is known to influence the substrate specificity of ADAM10 (Lipper, et al., 2023; Jouannet, et al., 2015). There might be a comparable mechanism for EGFR." These sentences are written in a confusing way- please clarify that you are raising the possibility that, based on which Tspan is bound to EGFR, this might direct specificity to a certain ligand (which is similar to the TspanC8s directing ADAM10 to a specific substrate)

Point-by-point responses:

Reviewer#1

Major comments:

1.1

1. [...]

This suggestion of additional experiments with an EGFR domain IV ectodomain mutant may represent a significant amount of work. The authors may consider tempering some conclusions instead, but experiments with such an EGFR mutant would significantly strengthen some key conclusions and the structural model.

We repeated the experiment shown in Fig. 2D with a deletion of the β -hairpin of EGFR and with the unmodified fusion protein in presence of EGF to address this (Fig. 4). Deletion of the β -hairpin results in faster dissociation of the EGFR-CD82 complex after cleavage with TEV protease. In contrast, prior binding of EGF to the complex does not affect dissociation, which is consistent with our model where domain IV interacts with CD82 and domain I and III are not affected and our MST experiments that detect the same binding affinity of EGF in presence and absence of CD82.

The deletion does not result in complete dissociation of the complex after complete cleavage after 2 h, which indicates either additional interactions of the transmembrane domains or an effect from micelle behaviour. The experiment with the β -hairpin deletion shows that dissociation speed is affected by this mutation (of the potential binding site of CD82) outside the micelle which supports that additional interactions apart from the two proteins being in the same micelle keep the complex from dissociating on SEC after TEV cleavage.

1.2

2. The strategy taken to express and purify an EGFR truncation comprised of the ectodomain and the transmembrane domain instead of the full-length EGFR may be necessary from a practical perspective. However, the kinase domain influences dimerization and also impacts ligand-binding affinity (e.g. PMID: 29997256). This should be discussed as a potential limitation of this study.

We added this to the limitations of the study in line 357-360 where we also discuss the possibility of intracellular factors effecting EGFR function possibly by binding to the intracellular tyrosine kinase domain.

1.3

3. The EGFR residues mutated to study the impact of EGF and CD82 binding (Y251A and R285S) appear to be in domain II, not domain IV of the ectodomain (as stated in lines 252-254 on page 8). An additional suggested reference for this could be PMID: 14732693.

Both mutations are indeed located in the dimerization arm in domain II and are intended to inhibit dimerization of EGFR. This was corrected in l. 303.

1.4

4. How does the addition of a His6 tag to EGF impact its interaction with EGFR? It would be helpful to

consider an assay to measure whether this tag has a significant impact of the interaction, such as perhaps comparing the ability of this EGFR-His6 to stimulate EGFR phosphorylation in cells compared to recombinant, commercially available EGF.

We cannot exclude that the His₆-tag possibly effects EGF binding affinity. However, in our experiments we are comparing the binding affinity to EGFR in absence and presence of CD82 using the same eGFP-EGF-His₆ construct (more precisely described in l. 274-275) and a change in binding affinity would affect the binding affinity to EGFR(I-TM) and EGFR(I-TM)-CD82 equally.

Minor comments:

1.5

1. The reference to the open, extended conformation of the EGFR ectodomain as the "active conformation" may be confusing, as EGFR activation typically refers to the formation of dimers with asymmetric arrangements of the kinase domains. Perhaps an alternative nomenclature can be used for the open, extended conformation of the EGFR ectodomain.

We changed the nomenclature from active conformation to extended conformation.

1.6

2. The structure of EGFR and CD82 shown in Figure 3 is a little difficult to see. The colors of some of the EGFR ectodomain regions (especially orange) are difficult to resolve from the red of the CD82. It would also be helpful to label each of these domains right on the image, rather than relying on the reader to match colors with statements in the figure caption.

We no longer fit structures into our density as it was pointed out that fitting is not reliable at a resolution of 15 Å (see comment 2.12).

Reviewer #2

2.1

Specific comments:

Line 46 "...the active conformation can form unstable dimers in the absence of ligand." Is that what is shown? Oligomers are present in the absence of ligand but I believe nothing about the nature or conformation of the oligomer in the absence of ligand is known.

Sako et al. (2000) describe a preformed EGFR dimer and therefore EGFR in what we inadvertently described as the "active" conformation. We realized that our nomenclature is confusing since EGFR is not active without binding ligand (also see comment 1.5) and therefore changed "active" to "extended" conformation throughout the manuscript.

2.2 and 2.3

Line 50 "In these self-activating oligomers, EGF-bound EGFRs flank a chain of unbound EGFRs (Liu et al. 2012)." I am puzzled by this statement, which is not reported or shown the cited reference (or elsewhere that I am aware of).

Line 51, I don't believe Needham et al. (2016) show direct evidence for "face-to-face" or "back-to-back" dimers. They simply show evidence for EGFR oligomers and make models. A more recent study of EGFR dimer clusters (Gonzalez-Magaldi et al. PNAS 2025) shows no regular interactions between EGFR dimer extracellular regions in EGFR clusters in vesicles studied by tomography.

The notion about self-activating oligomers from comments 2.2 and 2.3 were deleted from the introduction.

2.4

Line 64 "The structure of a CD19-CD81 fusion protein reveals interactions between the extracellular domains (Susa et al. 2020)." Susa et al. model interactions between CD19 and CD81 but no complex structure is determined.

The reference was corrected to Susa, et al. (2021). DOI: 10.1126/science.abd9836

2.5

Line 105 "likely represent two different glycosylation species" What is MW of CD82 and how many glycosylation sites does it have?

More information about N-glycosylation of CD82 at three sites in the LEL was added to the text (l. 110-112): CD82 is N-glycosylated at three sites located in the LEL and the two bands likely represent two different glycosylation species and have been observed as a double band in previous studies (Wang, et al., 2012; Odintsova, et al., 2000).

2.6

Does Figure 1E really show co-migration? Does a Western blot confirm that EGFR-His6 is present in earlier elution fractions?

We did not confirm the bands on the SDS-PAGE gel of the SEC by western blotting. We observed a band at the same height as we identified as EGFR in a previous small-scale purification. Because the complex was previously identified (Odintsova et al. 2000) and we did not achieve the desired purity we moved on to fuse EGFR(I-TM) and CD82 for which we see remaining complex after full cleavage via TEV protease (Fig. 2A). This experiment is now also better validated by repeating it with an EGFR mutant (See also comment 1.1).

2.7

It would be helpful to label figures 2B and 2C Coomassie and GFP, respectively.

We added labels underneath the gels in all figures where we show Coomassie staining and eGFP signal of the same gel.

2.8

Line 349 what is the transfection method/reagent?

Polyethyleneimine was used as transfection reagent and is now described in the materials and methods (l. 406).

2.9

Line 179 "This suggests that CD82 dissociated from EGFR (I-TM), even though they were still fused." What does this mean? Was the fusion protein cleaved? If so, is there evidence for the cleavage? If not, why is low-res density for CD82 not seen?

The fusion protein was not cleaved. However, we observe EGFR dimers in the absence of density for CD82 or micelles in 2D classes and in 3D. This observation suggests that EGFR and CD82 are split into separate micelles. Because of the length and flexibility (31 amino acids with GGS repeats and TEV cleavage site) of the linker the micelles are separated and not observed together in the 2D classes.

We cannot be certain that monomeric CD82 is present in our 3rd subgroup of 2D classes because of the small size of CD82 and presence of a second blurry density in some 2D classes (Fig.3A, "Micelle and CD82" 2D classes).

The text in line 190-193 was adjusted for clarity: "This suggests that CD82 dissociated from EGFR(I-TM)-dimer containing micelles (into a separated micelle), even though it remains fused to EGFR(I-TM)."

2.10

Lines 150. It is not stated, but I presume EGF was only added to one of the two samples prepared?

Yes, it was only added to one of the purifications. We adjusted the text in l. 159 for clarification: "Two samples of EGFR(I-TM)-CD82-StrepII₃ fusion protein were prepared, one with EGF-His₆ and one without."

2.11

Line 163 "The 2D classification of both samples resulted in three subgroups of particles. The first subgroup...was only observed for [the] sample containing EGF-His6". These statements are confusing. Were 3 different subgroups found for + vs. - EGF? Were similar subgroups found? It would help to be clear about which groups were found in + vs. - EGF samples.

The text was clarified and it was made explicit what 2D classes were observed in which data sets (l. 173-181): "A first subgroup, observed only in the presence of EGF, displayed a blurry micelle with a large, symmetric particle (**Fig. 3A**). 2D averages of this subgroup strongly resembled the structure of an EGF-bound EGFR dimer (Huang, et al., 2021). A second subgroup of 2D classes, present in both data sets, displayed a smaller, highly flexible averaged particle on top of a bigger micelle with at least two visible domains (**Fig. 3A**). This suggested, the presence of monomeric EGFR in these 2D classes (Lu, et al., 2012). A third and last subgroup of 2D classes, observed in both data sets, displayed a sharply resolved, slightly curved micelle containing an asymmetrically positioned particle of the expected size of a tetraspanin (**Fig. 3A**)."

2.12

Line 185 "Individual domains of the closed conformation of EGFR...were fitted into the density." From Figure 3 it looks like the individual domains were modeled independently despite there being no

evidence that the domains move independently in the absence of ligand and that even in the presence of ligand most interdomain movement occurs between domains II and III. The fits also look wholly unpersuasive-given the poor quality of the maps and the contortions of the EGFR domains, I see no reason at all to believe the model.

We agree that proteins cannot be modelled into the density with certainty. Therefore, we changed the figure to only the density and compare it to interactions observed in AlphaFold2 multimer predictions and further confirmed those with a following mutagenesis experiment (see comment 1.1).

2.13

Line 253 "...we repeated the F-SEC experiments with an EGFR double mutant, Y251 and R285S, which are located in the domain IV dimerization arm...". These amino acids are in the domain II dimerization arm.

This was corrected to domain II (l. 303).

2.14

Line 316 "EGFR can be activated by six other high and low affinity, such neuregulin and TGFalpha...". Neuregulin is not among the other six ligands that bind to EGFR.

Neuregulin was corrected to the low-affinity binder epiregulin (l. 371).

Reviewer #3

3.1

1. EGFR-CD82 refinement only had about 15k particles- is it possible to collect more data to improve the quality of the map? This would greatly improve the story. Are there any EGFR antibodies that you could use as fiducials to try to improve the map?

We tried to recover more particles via heterogeneous refinements. However subsequent ab initio modelling yielded poor densities. Further cleaning of particles yielded the same model based on a similar amount low number of particles as in the original density.

We tried to add EGFR nanobody 7D12 (PMID: 21520037, Schmitz et al. 2013) to the fusion protein of EGFR(III-TM)-CD82 to aid structure determination. However the nanobody we used bound domain III which did not help to address our issues with flexibility between EGFR(ECD) and the micelle, and eventually did not help achieving better resolution.

3.2

2. Line 178: "This suggests that CD82 dissociated from EGFR(I-TM), even though they were still fused" How do you think this is happening? Is the linker still present in your sample, or did you add TEV protease? If the linker was still present, are there any classes with two micelles close together (one

with EGFR/EGF, and one with CD82?) Please update the text to clarify- It's not clear to me how CD82 would be dissociating if it is fused

See comment 2.9

3.3

3. Add a final model figure describing your model in lines 266-283

A figure illustrating the overall mechanism was added as Fig. 7.

3.4

4. It is very interesting that CD82 inhibits EGFR dimerization. Is there any cell based evidence that overexpressing CD82 in cells attenuates EGFR signaling?

We are not aware of papers investigating this specifically. There is evidence that CD82 attenuates EGF induced morphological changes of HB2 cells and decreases internalization speed of EGFR (Odintsova et al. 2000)

3.5

5. Line 64: The CD19-CD81 structure also showed interactions between the TMs, and it seems (from what we know now) that most Tspans interact with their partners through both the extracellular and TM region- please update text to clarify. The reference for this should also be Susa et al, 2021 (Science paper, not the eLife paper).

The reference was corrected to Susa, et al. (2021).

3.6

6. Line 67: There are 6 members of the TspanC8 family shown to interact with ADAM10, not 10.

Corrected to six tetraspanins interacting with ADAM10 in l. 72.

3.7

7. Figure 2D: Using colors instead of grey scale would make this figure easier to quickly interpret for readers.

We added an experiment, repeating the assay with a mutant and in presence of EGF. We now use the colours to clearly distinguish the samples in Fig 2D and Fig. 4C-E and therefore would like to keep the colour gradients.

3.8

8. Line 149-150: Update to say "Two samples of EGFR(I-TM)- CD82-StrepII3 fusion protein were prepared, one with EGR, and one without."

We added the proposed clarification about the samples in l. 159.

3.9

9. Figure 3B: Please color EGF and EGFR different colors and add labels

Figure 3B only shows the resulting density without fitting. Figure S4 (previously S5) is now showing EGFR and EGF in different colours and labels were added.

3.10

10. Figure 3C: Please add labels for EGFR and the Tspan

The figure was changed as it was pointed out that fitting molecules at a 15 Å resolution is not precise enough. We moved the AlphaFold2 multimer prediction with the highest pTM score to the main figure and added labels to it.

3.11

11. Line 322: "Further experiments could focus on the combined effect of different tetraspanins and ligands on EGFR mechanics and signalling. A subgroup of tetraspanins is known to influence the substrate specificity of ADAM10 (Lipper, et al., 2023; Jouannet, et al., 2015). There might be a comparable mechanism for EGFR." These sentences are written in a confusing way- please clarify that you are raising the possibility that, based on which Tspan is bound to EGFR, this might direct specificity to a certain ligand (which is similar to the TspanC8s directing ADAM10 to a specific substrate)

The sentence was rephrased to (line 376-380): Further experiments could focus on the combined effect of different tetraspanins and ligands on EGFR mechanics and signalling. Possibly, the interacting tetraspanin has an effect on ligand specificity of EGFR in a comparable manner as substrate specificity of ADAM10 is impacted by the interacting tetraspanin (Lipper, et al., 2023; Jouannet, et al., 2015).

February 20, 2026

Re: Life Science Alliance manuscript #LSA-2025-03426-TR

Prof. Piet Gros
Utrecht University
Dept. of Chemistry
Crystal and Structural Chemistry
Utrecht 3584 CH
Netherlands

Dear Dr. Gros,

Thank you for submitting your revised manuscript entitled "Mechanistic implications on EGFR regulation by tetraspanin CD82" to Life Science Alliance. The manuscript has been seen by two of the original reviewers whose comments are appended below. While the reviewers continue to be overall positive about the work in terms of its suitability for Life Science Alliance, some important issues remain.

Reviewer 3 has reiterated their previous points on (1) your interpretation of disassociation of CD82 into a separate micelles taking into account the provided SEC trace and 2D classes and (2) clarifications on the Cryo-EM workflow and data.

Our general policy is that papers are considered through only one revision cycle; however we are open to one additional short round of revision. Given the importance of these points to your overall conclusions, a revised manuscript must address these concerns raised by the reviewer.

Please submit the final revision within one month, along with a letter that includes a point by point response to the reviewer's comments.

To upload the revised version of your manuscript, please log in to your account: <https://lsa.msubmit.net/cgi-bin/main.plex>
You will be guided to complete the submission of your revised manuscript and to fill in all necessary information.

B. MANUSCRIPT ORGANIZATION AND FORMATTING:

Sincerely,

Sarita Hebbbar, PhD
Scientific Editor
Life Science Alliance
<http://www.lsjournal.org>

Reviewer #1 (Comments to the Authors (Required)):

The revised manuscript addresses the comments from the review of the initial manuscript submission. In particular, new experiments that probe the dissociation of EGFR and CD82 with a mutant of EGFR in ectodomain IV are an important addition that strengthens the overall conclusions of the manuscript and the model presented. This new experiment also shows that the interaction of EGFR and CD82 are unlikely to be due to the two protein constructs being in the same micelle. Instead, these show that the interaction of EGFR and CD82 are likely mediated by the beta-hairpin in ectodomain IV of EGFR, supporting a specific interaction. Overall, this manuscript provides important evidence of a regulatory role of CD82 in gating the formation of EGFR dimers and ligand binding.

Reviewer #3 (Comments to the Authors (Required)):

Lamottke et al. present a nice manuscript on the interaction between EGFR and CD82. They describe protein engineering efforts to isolate a stable EGFR-CD82 complex, present a low-resolution structure of EGFR bound to CD82, and provide biochemical evidence that CD82 inhibits EGFR dimerization. Overall, I think the paper is well written, worthy of publication, and makes an important advance for the field. The authors have addressed most of my comments. However, I am not convinced the interpretation that CD82 is dissociating into a separate micelle upon EGF addition, even though they remained fused is correct, and I think it is worth investigating this further before final publication to ensure that is actually what is happening.

Even though the GS linker is reported as "long" (~31 amino acids), the flexible GS linker will dramatically increase the apparent affinity between EGFR-CD82 because it will keep the two proteins at a very high effective local concentration, promoting rapid rebinding and dramatically lowering apparent K_d . If CD82 were dissociating into a separate micelle, an additional DDM micelle adds about ~65 kDa of additional mass, so I would expect to see two peaks on the SEC trace (one for EGFR-CD82 in one micelle, and one for EGFR and CD82 in two separate micelles). However, I don't see any evidence of this in Fig S1. From past experience on another Tspan-partner complex, I also saw no density for the Tspan in a small data set collected on the Arctica, and I was worried that the Tspan dissociated, but the density only became obvious during larger collections on the Krios, which makes me wonder if this is just a resolution issue. Are there many 2D classes with two micelles tethered closely together? Because CD82 and EGFR are fused 1:1, I would assume if your hypothesis is correct that they are going into separate micelles, your 2D classes would have two micelles closely placed together if you increased the box size. That would support your hypothesis, but I don't see any of those classes included in your cryo-EM processing figure.

I am also curious about your cryo-EM data processing workflow. For the EGFR-CD82 complex without EGF, to my eye, it looks like the grey volume (56,303 particles) before TOPAZ picking actually had more features and possibly better resolution? It could just be because I am not looking at the actual maps, but I am wondering why the final 14.6Å map seems to look worse than that initial map.

For the EGF-EGFR-CD82 processing, I am also curious about the 3 ab initio models-- to me, none of them look like they are in the final dimerized extended conformation that you refine to 5.7Å. Which ab initio model was used for that refinement? You might have already tried this, but I am wondering if using more than three ab initio models might better separate out your particles since it seems like there are multiple conformational states in this data set. Did the blue ab initio model (75,519 particles), where it looks like the ectodomain is bent downwards, refine to anything interesting?

Reviewer #3 comments:

Even though the GS linker is reported as "long" (~31 amino acids), the flexible GS linker will dramatically increase the apparent affinity between EGFR-CD82 because it will keep the two proteins at a very high effective local concentration, promoting rapid rebinding and dramatically lowering apparent Kd. If CD82 were dissociating into a separate micelle, an additional DDM micelle adds about ~65 kDa of additional mass, so I would expect to see two peaks on the SEC trace (one for EGFR-CD82 in one micelle, and one for EGFR and CD82 in two separate micelles). However, I don't see any evidence of this in Fig S1.

The behaviour of two linked micelles on SEC column is not well studied. The observed broader, but single, peak would suggest a fast equilibrium between separated, but linked, and merged micelles on the column.

From past experience on another Tspan-partner complex, I also saw no density for the Tspan in a small data set collected on the Arctica, and I was worried that the Tspan dissociated, but the density only became obvious during larger collections on the Krios, which makes me wonder if this is just a resolution issue.

We tried local refinements with the maximum resolution set to 10 Å to avoid averaging out of any low resolution or flexible features. There are no additional densities at the micelle that might represent a tetraspanin. We added the result to the MS (Fig. S4B)

Are there many 2D classes with two micelles tethered closely together? Because CD82 and EGFR are fused 1:1, I would assume if your hypothesis is correct that they are going into separate micelles, your 2D classes would have two micelles closely placed together if you increased the box size. That would support your hypothesis, but I don't see any of those classes included in your cryo-EM processing figure.

There are indeed lots of 2D classes that show two micelles in close proximity. Even after several rounds of template picking these classes still occur. Because of the low resolution we did not select those classes for ab initio modelling which is why they are not shown in the processing figure. We added 14 2D classes of deselected particles as figure S4A.

I am also curious about your cryo-EM data processing workflow. For the EGFR-CD82 complex without EGF, to my eye, it looks like the grey volume (56,303 particles) before TOPAZ picking actually had more features and possibly better resolution? It could just be because I am not looking at the actual maps, but I am wondering why the final 14.6Å map seems to look worse than that initial map.

We were unsure if the features originated from overfitting and we did not see a complete micelle hinting towards preferred orientation of particles. We continued processing to improve those points but without success.

We instead focussed on the particles resulting of the TOPAZ cleaning because it yielded more promising looking 2D classes containing more particles compared with the original picking method and therefore more particles for 3D classification.

The difference is caused by our settings used for non-uniform refinement. We wanted to avoid overfitting or artefacts and therefore set the maximum resolution to 5 Å while it was not set for the volume of the original dataset.

For the EGF-EGFR-CD82 processing, I am also curious about the 3 ab initio models-- to me, none of them look like they are in the final dimerized extended conformation that you refine to 5.7Å. Which ab initio model was used for that refinement? You might have already tried this, but I am wondering if using more than three ab initio models might better separate out your particles since it seems like there are multiple conformational states in this data set. Did the blue ab initio model (75,519 particles), where it looks like the ectodomain is bent downwards, refine to anything interesting?

We clarified which models are used in the processing figure.

Indeed, the initial model used rather looks like the EGFR-CD82 complex (which is also present in this dataset) and the EGFR dimer only appeared clearly after several rounds of ab initio modelling and heterogeneous refinements. To clarify this, we added an additional 3D volume to the processing that shows the first observation of what we interpreted as EGFR dimer.

March 31, 2026

RE: Life Science Alliance Manuscript #LSA-2025-03426-TRR

Prof. Piet Gros
Utrecht University
Dept. of Chemistry
Crystal and Structural Chemistry
Utrecht 3584 CH
Netherlands

Dear Dr. Gros,

Thank you for submitting your revised manuscript entitled "Mechanistic implications on EGFR regulation by tetraspanin CD82". A few outstanding concerns in your manuscript were evaluated by one of the original reviewers. As you will read, Reviewer 3 has stated that your manuscript has addressed all their pending comments.

We would be happy to publish your paper in Life Science Alliance pending final revisions necessary to meet our formatting guidelines.

MANUSCRIPT ORGANIZATION AND FORMATTING:

To avoid unnecessary delays in the acceptance and publication of your paper, please read the following information carefully. Full guidelines are available on our Instructions for Authors page, <https://www.life-science-alliance.org/authors>

- We request you to reorganise your text to separate the Results from the Discussion section.
- When referring to results in Fig. 3D and Figure S5, please clearly state that these are AlphaFold predictions. For example on line 197, we request you to modify the sentence accordingly.
- Please provide a scale bar and size information for Figure 3C.
- Please confirm if Figure 5A and 5B are correctly described in the associated legend. In the present form the image heading and the description in the legend do not match.
- Please confirm if the same blot image has been used in (1) Figure 2B and Figure 4A and (2) Figure 2C and Figure 4B. If the same image has been utilised, then please indicate as such in the figure legends of both panels.
- You have the option of using Figure 7 as a Graphical Abstract in which case you would have to remove this figure file and upload it instead as a graphical abstract file, and edit the text accordingly.
- We encourage you to modify the title so that it is more informative.
- Please add a Running Title in our system.
- Please add the X and Bluesky handles of your host institute/organisation, as well as your own and/or one of the authors in our system.
- Please mark the corresponding author on the manuscript title page.
- Please add your main and supplementary figure legends to the main manuscript text after the references section.
- Please use the [10 author names et al.] format in your references (i.e., limit the author names to the first 10).
- Please add callouts for Figures S6A-D and S7A-B to your main manuscript text.
- Please remove legends from the supplementary figures and provide them in the main manuscript file.
- Please be sure that the authorship listing and order is correct

We welcome submissions of potential cover images for the issue of LSA in which your work would appear. If you have high

quality images associated with this work, please feel free to email these, with a caption, to the journal office.

LSA encourages authors to provide a 30-60 second video where the study is briefly explained. We will use these videos on social media to promote the published paper and the presenting author (for examples, see <https://docs.google.com/document/d/1-UWCfbE4pGcDdcgzcmiuJI2XMBJnxKYeqRvLLrLSo8s/edit?usp=sharing>). Corresponding or first-authors are welcome to submit the video. Please submit only one video per manuscript. The video can be emailed to contact@life-science-alliance.org

FINAL FILES:

The following items are required for acceptance.

The license to publish form must be signed before your manuscript can be sent to production. A link to the license to publish form will be available to the corresponding author only. Please take a moment to check your funder requirements.

Thank you for your attention to these final processing requirements. Please revise and format the manuscript and upload materials as soon as you are able.

Thank you for this interesting contribution to the literature. We look forward to publishing your paper in Life Science Alliance.

Sincerely,

Sarita Hebbbar, PhD
Scientific Editor
Life Science Alliance
<http://www.lsajournal.org>

Reviewer #3 (Comments to the Authors (Required)):

The authors have addressed my comments. Congratulations on the paper!

Dear Editors,

Please find the latest changes to the manuscript according to your comments. We hope that the manuscript is now suitable for publication.

Thank you for your efforts.

Kind regards,

Piet Gros

-We request you to reorganise your text to separate the Results from the Discussion section.

We removed all discussion items from the result section and connected them into a new discussion section.

-When referring to results in Fig. 3D and Figure S5, please clearly state that these are AlphaFold predictions. For example on line 197, we request you to modify the sentence accordingly.

We made sure the text and figure legends clearly state that the models were obtained from AlphaFold2 multimer.

-Please provide a scale bar and size information for Figure 3C.

All cryo EM volumes were flanked by scale bars that indicate width and height of the volumes.

-Please confirm if Figure 5A and 5B are correctly described in the associated legend. In the present form the image heading and the description in the legend do not match.

Protein names were switched in the figure legend, this was corrected.

-Please confirm if the same blot image has been used in (1) Figure 2B and Figure 4A and (2) Figure 2C and Figure 4B. If the same image has been utilised, then please indicate as such in the figure legends of both panels.

We show different lanes of the same gel in figure 2B,C and 4A,B. This is now explicitly indicated in the figure legend of Fig. 4a,b. We provided the uncropped gel as source data for further confirmation.

-You have the option of using Figure 7 as a Graphical Abstract in which case you would have to remove this figure file and upload it instead as a graphical abstract file, and edit the text accordingly.

We prefer not using Fig. 7 as a graphical abstract.

-We encourage you to modify the title so that it is more informative.

We changed the title to “Biochemical and structural data indicate mechanism of EGFR regulation by tetraspanin CD82”.

-Please add a Running Title in our system.

Binding mechanism of EGFR and CD82

-Please add the X and Bluesky handles of your host institute/organisation, as well as your own and/or one of the authors in our system.

Utrecht University:

Bluesky: utrechtuniversity.bsky.social

X: UU does no longer use X and has no active account

-Please mark the corresponding author on the manuscript title page.

The corresponding author is now indicated on the title page.

-Please add your main and supplementary figure legends to the main manuscript text after the references section.

The figure legends have been moved to after the reference section and the supplementary figure legends have been added.

-Please use the [10 author names et al.] format in your references (i.e., limit the author names to the first 10).

We adjusted the references to the suggested format.

-Please add callouts for Figures S6A-D and S7A-B to your main manuscript text.

We adjusted the callouts for figure S7 and S8 to S7A-D and S8A-B.

-Please remove legends from the supplementary figures and provide them in the main manuscript file.

We added the supplementary figure legends after the main figure legends.

-Please be sure that the authorship listing and order is correct

We are sure.

April 28, 2026

RE: Life Science Alliance Manuscript #LSA-2025-03426-TRRR

Prof. Piet Gros
Utrecht University
Dept. of Chemistry
Structural biochemistry
Utrecht 3584 CH
Netherlands

Dear Dr. Gros,

Thank you for submitting your Research Article entitled "BIOCHEMICAL AND STRUCTURAL DATA INDICATE MECHANISM OF EGFR REGULATION BY TETRASPANIN CD82" and fulfilling our formatting requests. We apologise for the delay in communicating our decision due to editor availability issues.

It is a pleasure to let you know that your manuscript is now accepted for publication in Life Science Alliance. Congratulations on this interesting work.

Your manuscript will now progress through copyediting and proofing. At the proofing stage, we highly encourage you to modify the title and we suggest the following, "A Potential mechanism for Tetraspanin (CD82)-mediated regulation of EGFR".

It is journal policy that authors provide original data upon request.

Your article will publish open access upon publication under a CC-BY license.

DISTRIBUTION OF MATERIALS:

Again, congratulations on a very nice paper. I hope you found the review process to be constructive and are pleased with how the manuscript was handled editorially. We look forward to future exciting submissions from your lab.

Sincerely,

Sarita Hebbar, PhD
Scientific Editor
Life Science Alliance
<http://www.lsajournal.org>